# Pacific Southern Ocean coccolithophore-estimated particulate inorganic carbon (PIC) versus satellite-derived PIC measurements

Mariem Saavedra-Pellitero[1], Karl-Heinz Baumann[2], Nuria Bachiller-Jareno[1], Harold Lovell[1], Nele Manon Vollmar[2,3], Elisa Malinverno[4]

[1]School of the Environment, Geography and Geosciences, University of Portsmouth, Portsmouth, PO1 3QL, United Kingdom
[2]Department of Geosciences, University of Bremen, 28334, Bremen, Germany
[3]NORCE Norwegian Research Centre AS, NORCE Climate & Environment, 5007, Bergen, Norway and Bjerknes Centre for Climate Research, Bergen, Norway
[4]Department of Geological Sciences and Geotechnologies, Milano-Bicocca University, 20126, Milan, Italy

*Correspondence to*: Mariem Saavedra-Pellitero (mariem.saavedra-pellitero@port.ac.uk)

## Abstract

Polar plankton communities are experiencing the impact of ocean acidification and global warming. Coccolithophores are the main type of calcifying phytoplankton in the Southern Ocean (SO) and they play a key role in the carbon cycle through the production of particulate organic, and inorganic carbon (PIC). However, in situ coccolithophore studies in the SO are sparse in space and time due to the harsh weather conditions. An alternative tool for monitoring PIC is the use of optical remote sensing, because coccolithophores account for most of the optical PIC backscattering in the sea. Here, we combine micropalaeontology and remote sensing to evaluate discrepancies between coccolithophore and satellite-derived PIC in the Pacific SO (in non-bloom conditions). Plankton samples were collected along two latitudinal transects: from New Zealand to Antarctica (December 2004-January 2005) and across the Drake Passage (February-March 2016). We compare PIC estimates derived from (1) Scanning Electron Microscope coccolith morphometric analyses and (2) MODIS-Aqua L2 and L3 PIC concentration values. In general, the coccolith-estimated PIC and satellite-derived PIC datasets show comparable trends in the Subantarctic and Polar Front Zones of both transects, with PIC-satellite values being generally higher than coccolith-derived PIC. However, satellite data availability was impacted by cloud cover in the SO. According to the coccolithophorid data, *Emiliania huxleyi* morphogroup B substantially contributes to the sea-surface PIC content south of the Subantarctic Front in both transects, whereas *E. huxleyi* type A, type A overcalcified, and other taxa (e.g. *Calcidiscus leptoporus*), only contribute to coccolithophore PIC in the northernmost stations. High satellite-derived PIC values south of the Polar Front, are not apparent in the coccolithophore data. We suggest that the high reflectance signal at the Antarctic Zone may instead relate to the presence of small biogenic opal particles (e.g. diatoms, silicoflagellates and/or small siliceous plankton) or other unknown highly reflective particles (such as *Phaeocystis* aggregations). Our results highlight the challenges presented by the lack of reliable

satellite data in some parts of the SO as well as the importance of in situ measurements and methodological accuracy when estimating PIC values. This work contributes to our understanding of coccolithophore PIC dynamics in the "data desert" of the vast Pacific SO, offering valuable insights into high-latitude phytoplankton and zooplankton communities.

## 1 Introduction

Coccolithophores are a major component of calcifying phytoplankton communities in the Southern Ocean (SO) (e.g. Saavedra-Pellitero et al., 2014; Saavedra-Pellitero et al., 2019; Malinverno et al., 2015; Charalampopoulou et al., 2016; Rigual Hernández et al., 2020a) and play an important and complex role in the carbon cycle through the production of particulate inorganic carbon (PIC) and particulate organic carbon (e.g. Rost and Riebesell, 2004; Salter et al., 2014). These haptophyte algae produce an external covering (coccosphere) of interlocking calcite platelets (coccoliths). This process decreases the alkalinity of surface waters, thereby reducing the uptake of $CO_2$ from the atmosphere into the surface ocean and thus acting in opposition to carbon sequestration by the biological carbon pump (Rost and Riebesell, 2004). Previous work has suggested that calcification during blooms of the coccolithophore *Emiliania huxleyi*, aka *Gephyrocapsa huxleyi* (Bendif et al., 2023), might alter the air-sea flux of $CO_2$ (e.g. Harlay et al., 2010; Shutler et al., 2013), although to date, the impact of this has mostly only been explored on a limited regional basis (e.g. Holligan et al., 1993; Robertson et al., 1994; Balch et al., 2016).

Since the early days of satellite-based color measurements of the oceans, large coccolithophore blooms have been visible as highly reflective regions in satellite images (e.g. Holligan et al., 1983). Coccolithophores, and their detached coccoliths, are strongly optically active and notably affect the optical budget of the surface ocean, and can thus be seen from space using satellite remote sensing (Smyth et al., 2002; Tyrrell and Taylor, 1996). Coccolithophores are responsible for most of the optical PIC backscatter in the ocean; the other, larger PIC particles associated with foraminifera and pteropods provide negligible backscatter per unit mass and therefore have minimal optical impact (Balch et al., 1996). In general, detached coccoliths account for 10-20% of the light backscattered from the sea under non-bloom conditions, whereas under bloom conditions it can be more than 90% (Balch et al., 1991; Balch et al., 1999). The strong scattering properties of the coccolithophores and the associated PIC lead to enhanced reflection in the entire visible spectrum (400-700 nm). Gordon et al. (2001) and Balch et al. (2005) developed algorithms to estimate the PIC concentration in the surface layer of the water column from the radiance emanating from the water. The relationship between inherent optical properties and the resultant light fields is well understood (e.g. Mitchell et al., 2017). The difficulty lies in understanding the combined effects of different in-water constituents on the inherent optical properties, and ultimately, the underwater light fields. While there have been many advances in this area (e.g. Babin et al., 2003a; Babin et al., 2003b; Devred et al., 2006), there will always be some uncertainty in calculating these relationships. For example, it has been shown that satellite ocean-color-based PIC estimates did not match in situ (ship-based)

observations and that satellite-derived PIC can be overestimated in Antarctic waters (e.g. Holligan et al., 2010; Trull et al., 2018). One potential source of error is that aquamarine waters characterized by high reflectance of light can also be caused by suspended sediment and even opal particles, such as fragments of diatom frustules (e.g. Broerse et al., 2003).

Satellite data has played a key role in showing the importance of the increasing *E. huxleyi* blooms in the world's oceans (e.g. Balch et al., 1991; Iida et al., 2002; Siegel et al., 2007; Neukermans et al., 2018; for further citations see the comprehensive review in Balch and Mitchell, 2023). This is relevant for monitoring changes at a global scale and to detect seasonal patterns as well as interannual variations (e.g. Smyth et al., 2004; Winter et al., 2014; Rigual-Hernández et al., 2020a) or trends, with the ultimate goal of feeding information into models for climate projections in the context of global warming and ocean acidification (e.g. Neukermans et al., 2018; Krumhardt et al., 2019). Recent concerns about climate change and ocean acidification pointed to *E. huxleyi* as a target cosmopolitan species to understand the biological response. Expansion or reduction of the biogeographic range, changes in coccolith calcification and preservation are possible responses that were observed in water and sediment samples. The high-latitude distribution of *E. huxleyi* has undergone a recent poleward expansion in both the northern (Rivero-Calle et al., 2015) and southern hemisphere (Cubillos et al., 2007; Winter et al., 2014). However, data from the SO is rather limited and there are currently not enough in situ measurements to unravel the complex dynamic relationships between *E. huxleyi* distribution and the frontal dynamics of the Antarctic Circumpolar Current (ACC). Significant zonal differences are shown in the relationship between coccolithophore data and ACC frontal positions across the different sectors of the SO (e.g. Saavedra-Pellitero et al., 2014), but no strong evidence of recent expansion on a circumpolar scale has been identified (Malinverno et al., 2015).

The band of high reflectance and elevated PIC waters observed in the SO between 30°- 60° S during Austral summer, known as "the Great Calcite Belt", has been linked to a region of increased seasonal abundance of coccolithophores (Balch et al., 2011; Balch et al., 2016). Comparisons of in situ and remote sensing measurements of PIC have been undertaken in the Atlantic and Indian sectors of the SO for coccolithophore bloom conditions (e.g. Balch et al., 2014; Balch et al., 2016; Poulton et al., 2011). Nonetheless, this type of comparison is very limited in specific areas of the globe (such as the vast Pacific sector of the SO) but also in non-bloom coccolithophore conditions. This is partially due to the fact that available coccolithophore measurements are sparse in space and time in the SO. Many of the subpolar studies focus on coccospheres, whilst there are scarce data on free coccoliths (Mohan et al., 2008).

Changes in the calcification of *E. huxleyi* coccoliths have been shown in sub-Antarctic waters, with different morphotypes representing the genotypic response to different water chemistry (Cubillos et al., 2007). Other studies (e.g. Beaufort et al., 2011; Horigome et al., 2014; Young et al., 2014) point to an environmental control on different calcification levels of *E. huxleyi*. Several estimates of coccolith-PIC exist; e.g. estimation of coccolith-mass from coccolith volume calculated from coccolith-size (Young and Ziveri, 2000; Beuvier et al., 2019) or estimation of coccolith-calcite mass through calibration of its

birefringence signal at the light microscope (Beaufort, 2005; Bollmann, 2014; Fuertes et al., 2014). Comparisons between
coccolith-estimated PIC and sea surface water scattering in the SO have targeted areas of coccolithophore blooms (Holligan
et al., 2010; Poulton et al., 2011; Balch et al., 2014; Oliver et al., 2023), but so far this has only occasionally been done for
non-bloom areas (e.g. Oliver et al., 2023).

Here, we focus one the contribution of *E. huxleyi* and other coccolithophore taxa to sea surface PIC along two latitudinal
transects across the ACC fronts: a New Zealand transect (December 2004-January 2005) and a Drake Passage transect
(February-March 2016). Coccosphere concentrations in the New Zealand transect were below $1.4 \times 10^5$ cells/L and in the Drake
Passage transect were below $1.5 \times 10^5$ cells/L, corresponding to non-bloom to outer bloom conditions (Poulton et al., 2011).
Our aims are: (1) to estimate the contribution of different coccolithophore taxa and morphotypes to PIC and (2) to compare
coccolith-based PIC estimates with satellite-derived PIC values in the Pacific SO.
**2 Study area: oceanographic setting and phytoplanktonic communities**
The SO is a high-nutrient, low-chlorophyll area in the Southern Hemisphere (e.g. De Baar et al., 1995) that connects all the
main oceans through the strong and eastward flowing ACC. In the SO, there are a number of oceanographic fronts characterized
by increased horizontal transport and rapid changes in water properties (Orsi et al., 1995; Klinck and Nowlin, 2001). The ACC
is bounded by the Subtropical Front (STF) in the north, which separates it from the warmer and saltier waters of the subtropics,
and its southern edge is marked by the Southern Boundary, which separates it from subpolar cold, silicate-rich waters (Orsi et
al., 1995). Although the ACC flow is mostly driven by the westerly winds, the position of the fronts varies spatially and
seasonally and it is also controlled by steep topographic features, such as oceanic plateaus or ridges (Gordon et al., 1978).
South of the STF, the Subantarctic Front (SAF) separates the Subantarctic Zone (SAZ) and the Polar Frontal Zone (PFZ) (Fig.
1). The location of the SAF is indicated by a strong thermal gradient and by the rapid descent of a salinity minimum associated
with the Antarctic Intermediate Water, from the surface in the PFZ (S<34) to depths greater than 300 m in the SAZ (S<34.20)
(Orsi et al., 1995; Whitworth, 1980). South of the SAF, the prominent Polar Front (PF) separates the PFZ and the Antarctic
Zone (AZ). The PF represents the northernmost extent of the 2°C isotherm at 200 m depth and corresponds to a 2°C gradient
in sea surface temperature (Orsi et al., 1995). The Southern ACC Front is characterized by temperatures below 0°C at the
minimum temperature in the sub-surface (<150 m) and above 1.8°C at the maximum temperature at depths >500 m (Orsi et
al., 1995). A more detailed description of the property indicators at each SO front can be found in Orsi et al. (1995).

Coccolithophores dominate the SO phytoplankton communities, especially in the SAZ, where they reach relatively high
numbers and diversity (e.g. Gravalosa et al., 2008; Saavedra-Pellitero et al., 2014; Malinverno et al., 2015; Charalampopoulou
et al., 2016; Saavedra-Pellitero et al., 2019; Rigual Hernández et al., 2020a). On the other hand, diatoms and other siliceous
microfossils dominate south of the PF (e.g. Saavedra-Pellitero et al., 2014; Malinverno et al., 2016; Cárdenas et al., 2018). The
coccolithophore abundance and diversity in the Drake Passage drastically drop from north to south, with the oceanographic
fronts appearing to act as ecological boundaries (Saavedra-Pellitero et al., 2019), whereas the total coccolithophore abundance
is highest in the PFZ south of New Zealand (Malinverno et al., 2015). Similar marked shifts at the SAF and PF in
coccolithophore number, community composition, and diversity occurring were also previously noted in other sectors of the
SO (e.g. Mohan et al., 2008; Gravalosa et al., 2008; Holligan et al., 2010; Saavedra-Pellitero et al., 2014; Balch et al., 2016;
Charalampopoulou et al., 2016) and are in accordance with previous observations in both transects (Malinverno et al., 2015;
Saavedra-Pellitero et al., 2019). In particular, the PF (Drake Passage) and the Southern ACC Front (New Zealand transect)
constitute natural sharp barriers marked by a clear drop in the number of *E. huxleyi*, which often is the only species found in
the PFZ and almost always occurs as B morphogroup (types B/C and O). Furthermore, a general southwards decreasing trend
in *E. huxleyi* mass, linked to a latitudinal trend from more calcified *E. huxleyi* (A morphogroup) to weakly calcified
morphotypes (B morphogroup), was already recorded across the Drake Passage (Saavedra-Pellitero et al., 2019).

**3 Materials and methods**
**3.1 Sampling considerations and morphometrics**
**3.1.1 The New Zealand transect**
Forty-two surface water samples were collected from the ship's pump of the *R/V Italica* (at ca. 3 m water depth) from 46.81°S
to 69.37°S during the XX Italian Expedition from New Zealand to Antarctica from 31st December 2004 to 6th January 2005,
(Fig. 1, Table 1). Details on sample locations, sampling volume, coccolithophore and coccolith counts can be found in
Malinverno et al. (2015).

We selected a total of 13 water samples for Scanning Electron Microscope (SEM, Vega Tescan at the University of Milano-
Bicocca) morphometric analyses of *E. huxleyi* covering the various biogeographic zones across the ACC (Fig. 1). For each
sample, 30-50 images of *E. huxleyi* free coccoliths and coccospheres were collected as encountered during filter scanning (377
images in total, Table 1S in Supplementary Material). Distal shield length and width, tube thickness, and number and thickness
of distal shield elements were manually measured using the ImageJ software (Schneider et al., 2012) in micrometers (µm)
using the scalebar of the SEM images (Fig. 2).
**3.1.2 The Drake Passage transect**
Nineteen water samples were collected on a transect at the western end of the Drake Passage (55.44°S to 61.75°S) during the
*Polarstern* Expedition PS97 from 24th February 2016 to 5th March 2016 (Fig. 1, Table 1). These selected plankton samples
were obtained using a rosette sampler with 24×12 L Niskin bottles (Ocean Test Equipment Inc.) attached to a CTD Seabird
SBE911plus device (Lamy, 2016). The bottles were fired by a SBE32 carousel and just the shallowest samples, from 5, 10 and
20 m water depth, were considered in this work. Details on sample locations, sampling volume, coccolithophore assemblages
and coccospheres/L can be found in Saavedra-Pellitero et al. (2019).

A total of 203 images of *E. huxleyi* coccospheres were taken from the samples in the Drake Passage while scanning the filters
within another SEM (Zeiss DSM 940A at the Geosciences Faculty, University of Bremen; Table 2S in Supplementary
Material). Coccoliths were measured using the Coccobiom2 macro (Young, 2015) in the software program Fiji, an image
processing package based on ImageJ (Schindelin et al., 2012). Measurements were made in µm, based on the scale bar of the
SEM images. Note that they were scaled to 100% with a Coccobiom2 SEM calibration of 1.09 and the specific magnification.
**3.2 Coccolithophore PIC estimates**
Species-specific coccolith-PIC (in pmol) was estimated following the volume calculation of Young and Ziveri (2000)

$PIC = (2.7 \times Ks \times L^3) \div 100$      [equation 1]

where:
2.7 = density of calcite;
*Ks* = species-specific shape factors, as provided by Young and Ziveri (2000) and modified for *E. huxleyi* according to the
degree of calcification obtained for each morphotype as compiled by Vollmar et al. (2022) (further details in Table 2);
*L* = coccolith mean length from measurements in the case of *E. huxleyi*. For minor species, we considered the averaged
coccolith length provided by Young and Ziveri (2000);
100 = molecular weight of calcite.

Measurements of the distal shield diameters of *Calcidiscus leptoporus*, the second most abundant species that is significantly
larger and much more massive than *E. huxleyi,* were made on different samples offshore of New Zealand, corresponding to
the highest abundances of this taxa (Table 2 and Table 3S in Supplementary Material). The importance of own size
measurements for the determination of species-dependent coccolith PIC has been clearly emphasized (Baumann, 2004). The
coccolith-PIC contribution for each sample was calculated by applying the obtained species-specific calcite quota to the
abundances of species and morphotype (i.e., coccospheres/L) from Malinverno et al. (2015) and Saavedra-Pellitero et al.
(2019) (Tables 1 and 2). In the New Zealand transect, the single / double coccolith layers were considered in the estimates
(Table 1S in Supplementary Material), while in the Drake Passage transect, where this information was not available, an
average was considered based on our own observations (Table 2 and Table 4S in Supplementary Material). Additionally,
detached coccoliths/L were considered for the PIC estimates in the New Zealand transect (Malinverno et al., 2015). To estimate
the number of coccoliths per coccosphere we counted the visible placoliths (half coccosphere) and multiplied by two (e.g.
Table 4S).
We also calculated the relative tube width in *E. huxleyi* as a size-independent index to estimate the degree of calcification in
this taxa following Young et al. (2014) (Fig. 2):

Relative tube width= (2×tube width)÷coccolith width                [equation 2]

Note that because the relative tube width is a ratio, it is dimensionless and it should be size-independent (Young et al., 2014).

## 3.3 Coccolith-estimated PIC errors

There are sources of errors and uncertainties linked to the approach chosen to estimate the coccolith PIC. To assess the
reproducibility of the measurements, two different coccoliths were measured 50 times each. The standard deviation (SD) for
the coccolith length was 0.014 and 0.017 μm and the standard error 0.002 μm in both cases. Coccolith volume estimates are
likely to contain errors around 40-50% according to Young and Ziveri (2000), so we assumed the largest potential error and
added a 50% error bars to our plots, although we note that measuring the actual size range in the sample can reduce this error
to about 5-10% in length and 15-30% in volume.

## 3.4 Satellite-derived PIC and chlorophyll a data processing

To compare the coccolith-estimated PIC with satellite-derived values, PIC concentration in mol m$^{-3}$ was obtained from the
MODIS-Aqua Level (L) 2 and L3 products (NASA Goddard Space Flight Center, Ocean Ecology Laboratory, Ocean Biology
Processing Group, 2022a). To encompass the broad range of PIC concentrations observed in the global ocean, a combination
of two independent approaches is used to calculate the backscattering coefficient for PIC (the description of the algorithm can
be found in NASA Ocean Biology Processing Group, 2023; for further details see also Balch and Mitchel, 2023). The Ocean
Biology Processing Group (OBPG) validates MODIS-Aqua PIC retrievals against in situ measurements, which results in a
mean bias of ± 0.31623 and a mean absolute error (MAE) of ± 3.91664 (both values calculated based on log10 transformation
to the PIC values) (https://oceancolor.gsfc.nasa.gov/data/reprocessing/r2022/aqua/). These metrics indicate the degree of
accuracy and potential bias in the satellite-derived estimates compared to direct observations.

MODIS-Aqua L2 scenes encompassing both the sampling period and the geographical extent of each transect were
downloaded from NASA's Ocean Colour L1 and L2 browser (https://oceancolor.gsfc.nasa.gov/cgi/browse.pl). The
downloaded MODIS L2 scenes corresponded to swaths covering at least 50% of the study area and included more than one

daily scene. Table 3 summarizes the number of downloaded scenes as well as their time coverage. To obtain satellite-derived PIC concentrations for comparison with coccolith-estimated PIC at each sample location, the mean of a 5x5 window centered on the measurement location (Bailey and Werdell, 2006) was extracted from the downloaded scenes using the SNAP 9.0.0 pixel extraction tool (European Space Agency (ESA), 2022). This tool provides basic statistics, such as the number of pixels (N) contributing to each mean value and the SD of these pixel values, allowing the homogeneity of the extraction point to be assessed. Pixels flagged with atmospheric correction failure (ATMFAIL) or very low water-leaving radiance (LOWLW) were excluded from the extraction. To ensure statistical confidence in the retrieved values, all PIC mean values resulting from the aggregation of 12 or fewer N within the 5x5 window were discarded (Bailey and Werdell, 2006). Duplicate daily mean PIC values (i.e. PIC values for a measuring location extracted from more than one scene captured on the same day) and their corresponding SD were then weighted according to their uncertainties (Bevington, 1969) to give more prominence to measurements with a lower SD, which are generally considered to be more reliable. However, daily mean values with SD equal to zero were used directly as the result, since the value with zero SD suggests homogeneity.

Due to high cloud cover and other conditions that interfere with the detection of water-leaving radiances (NASA Ocean Biology Processing Group, 2023), daily PIC grids yielded a high number of missed observations, or gaps, which prevented us from acquiring daily satellite-derived PIC values of the sampling dates for most sample locations in both transects (Figs. 1S and 2S in Supplementary Material show the availability of MODIS-Aqua L2 PIC values across stations over the sampling period). This data scarcity made it impossible to use a time window of 24 h to determine coincidence between coccolith-estimated PIC and satellite-derived PIC. Therefore, to increase data availability, we (1) extended the satellite period to seven days before and after sampling dates (see Table 3 for specific dates) and extracted the PIC for all sample locations, regardless of their sampling date. We deliberately chose that time range considering that *E. huxleyi* can double its numbers in two or three days without accounting for grazing by zooplankton (based on studies in the North Atlantic; Holligan et al., 1993), ensuring no drastic changes from non-coccolithophore bloom to bloom conditions. We then generated a mean PIC value for each location by aggregating the available daily means over the full period to explore the latitudinal variation of this variable; and (2) also obtained monthly (Figs. 3S and 4S in Supplementary Material) and 8-daily (going forward, referred to as weekly) satellite-derived PIC concentrations (mol m$^{-3}$) from the MODIS-Aqua L-3 product (NASA Goddard Space Flight Center, Ocean Ecology Laboratory, Ocean Biology Processing Group, 2022b). This allowed us to have additional satellite-derived PIC values to compare to the coccolith-estimated PIC in the study area. Images encompassing both the sampling period and the geographical extent of each transect, were acquired from NASA's Ocean Color Level 3 and 4 Browser (https://oceancolor.gsfc.nasa.gov/l3/) as 4 km cell size gridded files in NetCDF file format. Table 3 summarizes the number of downloaded scenes as well as their time coverage. The L3 extracted values corresponded to the PIC concentration of the grid cell enclosing the sample location. As per L2 data extraction, PIC concentrations for all sample locations were acquired from all available monthly and weekly scenes. MODIS-Aqua L2 chlorophyll a concentration in mg m$^{-3}$ were also extracted

and processed as an indicator of the presence of diatoms and other phytoplanktonic groups. The algorithm used to calculate
chlorophyll a is documented by Werdell et al. (2023).
**4 Results**
**4.1 Coccolith-estimated PIC versus satellite-derived PIC**
*Emiliania huxleyi* is the dominant species in the coccolithophore assemblage of the Pacific SO (Malinverno et al., 2015;
Saavedra-Pellitero et al., 2019) with abundances of $1.4x10^5$ coccospheres/L (at station TR033) south of the SAF in the New
Zealand transect and $1.5x10^5$ coccospheres/L (at station PS97/034-2) in the Drake Passage SAZ and it is also the main
contributor to sea-surface PIC (Figs. 3 and 4). *Calcidiscus leptoporus* (mostly the intermediate-sized form) is the second most
abundant species and makes significant contributions to the coccolithophore PIC at certain locations (up to $1.4x10^4$ cells/L in
the New Zealand transect and $1.4x10^3$ cells/L in the Drake Passage, Figs. 3 and 4) (Malinverno et al., 2015; Saavedra-Pellitero
et al., 2019). *Calcidiscus leptoporus* generally represents on average 20.2% of the total coccolithophore PIC in the New
Zealand transect and 5.3% in the Drake Passage, but can occasionally reach maximum PIC contributions of 68.3% (at station
TR008, in the SAZ) and of 31.1% (at station PS97/017-1, in the SAZ) (Fig. 5).
A minor contribution from less abundant or rare species is found in the northern SAZ of both transects, where diversity is
higher (for species list see Malinverno et al., 2015; Saavedra-Pellitero et al., 2019), with a poleward decreasing trend and
almost no contribution south of the SAF (Fig. 5). *Emiliania huxleyi* is responsible for almost all of the coccolith-estimated PIC
in the PFZ, but its contribution decreases at the PF (in the Drake Passage) and Southern ACC Front (in the New Zealand
transect, ca. 63.7°S) and further south. Daily, weekly and monthly satellite (MODIS-Aqua L2)-derived PIC at the sampling
locations are generally higher than coccolith-estimated PIC in both transects; this difference is larger in the Drake Passage
(Fig. 4) than in the New Zealand transect (Fig. 3). There are discrepancies in absolute values (on top of the already inherent
variations in the weekly compared to the monthly PIC estimates and the limited availability of L2 data). These are particularly
obvious at the PF (ca. 60°S in the Drake Passage) or to the south of it (ca. 62.5°S in the New Zealand transect), where the
satellite-derived and coccolith-estimated PIC become decoupled, characterized by high reflectance but no coccolithophores in
the AZ.

**4.2 Morphometries and mass estimates of *Emiliania huxleyi***
*Emiliania huxleyi* consist of different morphotypes that show a different and partly overlapping distribution along both
latitudinal transects (Malinverno et al., 2015; Saavedra-Pellitero et al., 2019). Type A is mostly restricted to the northern SAZ,
but it is occasionally present in the PFZ in the Drake Passage (Figs. 3, 4) and it is the only type within morphogroup A in this
study. Morphotypes belonging to the *E. huxleyi* morphogroup B (which includes morphotypes B, B/C, C and O) are present in

the SAZ and the PFZ, but they disappear south of the PF. Morphometric measurements on coccoliths of *E. huxleyi* from the selected samples show that the length of types A, B/C-C and O overlap in both transects (Fig. 6). In the Drake Passage, coccolith lengths range from 2.86 to 3.96 ± 0.43 μm (unless specified, ± refers to the SD from now on) with a mean average of 3.49 ± 0.33 μm for A type (including normal and overcalcified specimens), 2.87 to 4.11 ± 0.45 μm for B type, 2.20 to 3.98 ± 0.37 μm for B/C-C types, 2.42 to 4.16 ± 0.41 μm for O type, and an average of 2.98 ± 0.40 μm for morphogroup B. In the New Zealand transect, maximum lengths range from 2.25 to 3.59 μm, with an average of 2.95 ± 0.28 μm for *E. huxleyi* type A, 1.95 to 3.62 ± 0.33 μm for B/C-C types, 2.07 to 4.14 ± 0.36 μm for type O, and an average of 2.87 ± 0.35 μm for morphogroup B.

Figure 6 provides a latitudinal overview of morphometric data compared to the (averaged) degree of calcification (indicated by the dimensionless relative tube width index; Young et al., 2014). In the New Zealand transect there are no significant changes in coccolith lengths except for a wide scatter of values characterizing the size class distribution of each sample. This feature reflects the large variability in coccolith size as observed on coccoliths from a single coccosphere (Fig. 2e). However, in the Drake Passage transect, *E. huxleyi* coccoliths are notably larger offshore of Chile (Fig. 6a).

*Emiliania huxleyi* masses calculated in the New Zealand transect range from 0.61 to 2.93 pg with an average of 1.47 ± 0.46 pg per coccolith belonging to the morphogroup A, and from 0.36 to 2.86 pg, with an average of 1.15 ± 0.43 pg per placolith from morphogroup B (Fig. 3e). In the Drake Passage the masses per coccolith for morphogroup A are almost double than in the New Zealand transect, varying between 1.39 pg and 6.26 pg, with an average of 3.00 ± 1.19 pg. The placolith masses in morphogroup B range from 0.57 to 3.75 pg with a mean of 1.44 ± 0.62 pg across the Drake Passage (Fig. 4e). The coccolith-estimated PICs for just the species *E. huxleyi* are generally lower in the New Zealand transect (average morphogroup A: 0.021 ± 0.010 pmol and B: 0.013 ± 0.006 pmol, considering 50% potential error) than in the in the Drake Passage (average morphogroup A: 0.034 ± 0.017 pmol and B: 0.014 ± 0.007 pmol -error-).

We observed that some coccoliths are clearly overcalcified (see Fig. 6), with a thick inner tube (up to 0.76 μm in sample PS97/018-1) that extends into the central area. Specimens belonging to the morphogroup A show a higher degree of calcification than those belonging to morphogroup B, resulting not only in a thicker inner tube but also in thicker distal shield T-elements. The overcalcified coccospheres co-occur with normally-calcified ones but they are restricted to the northernmost samples (Fig. 6). The relative tube width (as an index for calcification), calculated using equation 2, varies from 0.10 to 0.28 ± 0.04 in morphogroup A and from 0.07 to 0.21 ± 0.03 in B for the New Zealand transect. Values are higher in the Drake Passage, ranging from 0.05 to 0.50 ± 0.12 for *E. huxleyi* morphogroup A, and from 0.02 to 0.22 ± 0.04 for morphogroup B. The degree of calcification is highly variable within each sample of the New Zealand transect (Fig. 3d), but overcalcified specimens (relative tube width >0.23), typically represented by type A, only occur in the northernmost samples (Fig. 6b). The averaged relative tube width index shows increased values not only in the SAZ offshore of New Zealand, but also around 54°S

and in the PFZ (Figs. 3d, 6b), which points to a certain degree of variation in the calcification within morphotypes BC/C and
O. A more marked N-S decrease in the relative tube width index values is observed in the Drake Passage, with notably higher
values offshore of Chile (Figs. 4d and 6a), where relatively large and heavily calcified type A coccospheres are present.

## 5. Discussion

### 5.1 PIC variability in the SAZ and PFZ

In the studied transects, even with the limited data available, the coccolith-estimated PIC and the satellite-derived PIC show a
comparable trend in the SAZ and PFZ, but there is a strong discrepancy in the AZ (Fig. 5S in the Supplementary Material).
The fact that satellite-derived PIC is generally higher than coccolith-estimated PIC in the SAZ and PFZ (Figs. 3, 4 and 5S in
the Supplementary Material) could be due to an underestimation of the species specific coccolith-estimated PIC. The potential
assumptions linked to the coccolith-estimated PIC, including shape factors (Ks), average coccolith length (L), number of
coccoliths per coccosphere, and/or number of coccolith layers per cell (Table 2), have associated uncertainties. Even if we
tried to minimize these by measuring the actual coccolith size range and counting coccoliths per coccosphere (instead of using
assumed values), the overall error can still add up to $\pm$ 50% (Young and Ziveri, 2000). Additionally, the fact that the difference
between satellite-derived PIC and coccolith-estimated PIC in the Drake Passage transect is larger than in the New Zealand
transect can be also in part attributed to the fact that detached coccoliths (in addition to coccospheres) were only considered in
the estimates for the New Zealand transect.

Given that *E. huxleyi* is the dominant species and the main contributor to coccolith-estimated PIC in the SAZ and PFZ of both
transects (Fig. 5), we focused on its abundance, morphotype distribution and calcite weight to assess potential PIC
discrepancies. Coccolith-estimated PIC for *E. huxleyi* are generally in agreement with the calcite content per coccolith obtained
by Poulton et al. (2011) along the Patagonian Shelf and by Rigual Hernández et al. (2020a) in the Australian and New Zealand
sectors of the SO (see Tables 1 in those papers). Our *E. huxleyi* PIC estimates seem generally higher than the estimates by
Charalampopoulou et al. (2016) off southern Chile (0.015 pmol per coccolith) and across the rest of Drake Passage (< 0.009
pmol). On the other hand, our values are slightly lower than those obtained through the birefringence method SYRACO, an
automated system of coccolith recognition (SYstème de Reconnaissance Automatique de COccolithes) in the same latitudinal
range (e.g. Beaufort et al., 2011), and notably lower than Saavedra-Pellitero et al. (2019), who used circularly polarized light
plus the C-Calcita software developed by Fuertes et al. (2014) across the Drake Passage (Fig. 7a). In order to explore this
difference, we calculated PIC in the Drake Passage using the Saavedra-Pellitero et al. (2019) mass estimates for the same
samples, considering an average mass of 4.64 $\pm$ 2.53 pmol for *E. huxleyi* (n = 796), but without distinguishing different
morphotypes (Fig. 7c). The mass per coccolith of *E. huxleyi* using C-Calcita in the Drake Passage is 2.8 times higher than in
this study (mean of 1.66 $\pm$ 0.91 pg here). We then extrapolated the potential contribution of the rest of the coccolithophore

taxa using this factor (i.e., multiplying by 2.8 the *C. leptoporus* and minor species PIC values calculated in this study) (Fig. 7c). Both N-S coccolith mass and PIC trends mirror each other, but the C-Calcita-derived PICs tend to overestimate satellite-derived PIC values, except in a couple of locations. This can be attributed to the calibration of the coccolith thickness within the software C-Calcita, which has been improved in recent years with the use of a calcite wedge instead of a calcareous spine (e.g. Guitián et al., 2022).

The generally higher satellite-derived PIC numbers compared to the coccolith-estimated PIC values in the SAZ and PFZ (Figs. 3, 4, and 5S in Supplementary Material) could be also due to the presence of other carbonate-forming organisms (and/or their fragments); for instance, foraminifera, can contribute to a significant fraction of the total PIC in the SO south of Australia, especially between 55-60°S (Trull et al., 2018). We do not have data for the Drake Passage, but planktonic foraminifera were observed in the filter samples across the New Zealand transect, showing increased abundance (together with the tintinnid species *Codonellopsis pusilla*) in the PFZ (see Malinverno et al., 2016 for further details). Although foraminifera and other hard-shelled micro-zooplankton PIC particles provide negligible backscatter per unit mass (Balch et al., 1996), they can be a source of error in the PIC volume calculation when considering only coccolithophores. Assessing the significance of carbonate-forming organisms relative to other taxa in the SO is an important topic, but falls beyond the scope of this paper.

Observed discrepancies between satellite-derived PIC and coccolith-estimated PIC values can arise from a combination of several factors related to the PIC algorithm sensitivities and limitations (Mitchell et al., 2017; Balch and Mitchell, 2023; NASA Ocean Biology Processing Group, 2023), differences in spatial and temporal resolution (Table 3), and environmental factors (e.g. turbidity or other particulate matter that can affect the accuracy of satellite-derived PIC estimates). MODIS-derived L2 PIC data was limited due to the cloudy skies of the SO during the sampling period (see Figs. 1S and 2S in the Supplementary Material). To mitigate the impact of these data gaps in our analysis, we extended the time window for data extraction to several days and computed the mean for each location, whilst also using L3 products. This approach, while necessary, could obscure potential variability at shorter temporal scales and create discrepancies when comparing with sample measurements taken on specific days. The fact that the overall trends are comparable in the New Zealand and Drake Passage transects (Fig. 5S in the Supplementary Material), could also suggest a satellite bias linked to the algorithm. We are aware that the MODIS-Aqua Ocean Color was re-processed in 2022 to incorporate updates in instrument calibration, new ancillary sources and algorithm improvements (NASA Ocean Biology Processing Group, 2023a), but the validation of the PIC measurements was based on a low number of in-situ measurements compared to other products (e.g. 1347 in situ measurements for chlorophyll a and just 42 for PIC, all of them in the Atlantic Ocean; NASA Ocean Biology Processing Group2023b). The differences in PIC could also be due to the fact that we are comparing in situ values to weekly and monthly averages, as well as also smoothing data by considering averaged values when estimating coccolith-estimated PIC (especially length and number of coccoliths per coccosphere). In addition, sampling at slightly different times of the year may also have an influence on the PIC values determined (Rigual Hernández et al., 2018; Rigual-Hernández et al., 2020a, b).

In general, we find a different pattern to that described in Balch et al. (2014), who determined coccolith-quotas in the center of a coccolithophore bloom in the Patagonian Shelf (Atlantic Ocean) ranging from 0.008 to 0.017 pg per coccolith by comparing automated coccolith-counts with coccolith-estimated PIC. In the context of other observations, the coccolith quotas calculated by Balch et al. (2014) are relatively low and show a much greater variation within a limited region. Considering the differences in the two SO transects studied here, which were sampled 11 years apart, we could assume that with our approach, the surface coccolith-estimated PIC (up to 20 m water depth) underestimate satellite-derived PIC concentrations in the SAZ and PFZ. This discrepancy is evident in our data, where coccolith-estimated PIC concentrations calculated using different methodologies, such as C-Calcita, exceed those obtained from the satellite data. This indicates that there is still a need for improved precision in coccolith-estimated PIC concentration methods. Therefore, it is crucial to refine existing methods and develop new algorithms to enhance accuracy. Additionally, the limited number of in situ data points used for calibrating satellite algorithms in the SO could contribute to these discrepancies, highlighting the importance of expanding in situ datasets for better validation and calibration of remote sensing data.

## 5.2 Assessing potential biases in PIC estimates for the AZ

In the AZ (south of about 62.5°S in the New Zealand transect and about 60°S in the Drake Passage), high reflectance is detected by remote sensing but is not associated with a coccolithophore bloom (Figs. 3, 4 and S5). Concentrations of *E. huxleyi*, which show maximum numbers in the PFZ at the New Zealand transect and moderate values in the Drake Passage, drop southward of this location at the Southern ACC Front and the PF (Malinverno et al., 2016). Satellite data show the different impact of ACC fronts on the distribution of *E. huxleyi* (Holligan et al., 2010): in the Drake Passage, where the fronts are strictly constrained by topography, *E. huxleyi* is bounded by the PF to the south (Saavedra-Pellitero et al., 2019), while in the eastern Scotia Sea, where the ACC fronts are broadly separated, *E. huxleyi* spreads between the PF and the Southern ACC Front (Holligan et al., 2010; Poulton et al., 2011; Poulton et al., 2013). This pattern also emerges from the compilation by Malinverno et al. (2016), which shows that the Southern ACC Front marks the southern boundary in different SO sectors.

Occasional occurrences of *E. huxleyi* south of the Southern ACC Front have been documented south of Tasmania and in the Weddell sea in certain years by conventional micropalaeontological observations (e.g. Winter et al., 1999; Cubillos et al., 2007) as well as in the Australian sector of the SO and in the Scotia Sea using surface reflectance data only (Holligan et al., 2010; Winter et al., 2014). However, in our study, *E. huxleyi* is constrained by the Southern ACC Front corresponding to a maximum sea surface temperature of 1°C in the New Zealand transect.

The magnitude and spectral characteristics of water-leaving radiance detected by satellites are influenced by the inherent properties of the optically active constituents. These include: (1) light scattering by PIC, other biogenic particles or lithogenic material (e.g. Bi et al., 2023) as well as (2) light absorption by phytoplankton biomass (i.e., chlorophyll a concentration) and dissolved organic matter (e.g. Reynolds et al., 2001; Ferreira et al., 2009). The strong correlation between high values of water-

leaving radiance and high *E. huxleyi* PIC concentrations has been successfully proved in bloom areas (e.g. Gordon et al., 1988;
Balch et al., 2005; Holligan et al., 2010; Balch et al., 2011; Balch et al., 2014; Balch and Mitchell, 2023; Oliver et al., 2023).
However, not all bright waters are caused by *E. huxleyi* blooms, as shown by Broerse et al. (2003) in the Bering Sea, Balch et
al. (2007) in the Gulf of Maine, and Daniels et al. (2012) in the Bay of Biscay. Suspended particles, which include either
reworked coccoliths, lithogenic material or empty diatom frustules, could be responsible for high values of water-leaving
radiance, at least in nearshore regions (Broerse et al., 2003; Balch and Mitchell, 2023).

The occurrence of bright waters along the studied transects should theoretically be constrained by the position of the PF/
Southern ACC Front. Malinverno et al. (2015; 2016) showed a significant shift in the community composition from carbonate
to silica-dominated microfossils in the New Zealand transect at the Southern ACC Front, with diatoms being the most abundant
mineralized phytoplankton group in the transect (Fig. 3k). Coccolithophores disappear south of the Southern ACC Front, and
the composition of the siliceous phytoplankton changes from a dominance of large diatoms (*Fragilariopsis kerguelensis*) in
the north to a dominance of small diatoms (such as the cold adapted *Fragilariopsis cylindrus*) in the south, with a notable
increase in spiny silicoflagellates (e.g. *Stephanocha speculum* var. *coronata*) and small siliceous plankton (Parmales,
Archaeomonads) (Malinverno et al., 2016) coincident with high values of chlorophyll a in the AZ (Figs. 3l, 8).  Extant diatoms
have not yet been studied in the exact same water samples collected during PS97 Expedition. However, the abundance of
subfossil diatoms in surface sediments in the Drake Passage shows an increase south of the PF, along with an increase in the
relative abundance of siliciclastics, biogenic opal (Cárdenas et al., 2018). This contrasts with the relatively low satellite-derived
chlorophyll a concentration in the AZ (Fig. 4k), but this only due to the very limited number of daily L2 data available.
*Fragilariopsis kerguelensis* appears to dominate up to the Southern ACC Front, and *F. cylindrus* is found south of this front,
in colder waters of the Drake Passage (Cárdenas et al., 2018).

Different alternatives have been suggested for the high reflectance in the AZ of the SO, such as microbubbles (mostly during
storms), floating loose ice, high concentrations of other particulate matter such as glacial flour (especially close to the Antarctic
continent) or *Phaeocystis* blooms (Balch et al., 2011; Balch, 2018; Balch and Mitchell, 2023). Our observations do not allow
us to comprehensively determine the potential causes of this high reflectance, but we note that a high abundance of small opal
biogenic particles, such as small-size diatoms, silicoflagellates and siliceous plankton observed (as well as their fragments)
would be consistent with the observed high scattering of these waters at least in the New Zealand transect (Figs. 1, 3, 4, 3S
and 4S in Supplementary Material), even though opal particles have a much lower refractive index than calcite (Balch, 2009;
Costello et al., 1995).

**5.3 *Emiliania huxleyi* morphotypes**
*Emiliania huxleyi* morphometric dataset reveals that type A overcalcified morphotype is highly distinct from the other
morphotypes (Fig. 6). This morphotype has also been previously observed in the coastal waters of the eastern South Pacific

and in the open ocean (Beaufort et al., 2011; Von Dassow et al., 2018; Saavedra-Pellitero et al., 2019). However, it should be noted that type A overcalcified in this work includes the moderately calcified, robustly calcified and extremely heavily calcified A morphotypes described by Diaz-Rosas et al. (2021). Coccospheres of *E. huxleyi* classified by Diaz-Rosas et al. (2021) as extremely heavily calcified R/hyper-calcified and/or A-CC morphotypes (with complete overgrowth of the coccolith central area but without fusion of distal shield elements) occasionally occurred offshore of Chile in samples closest to the coastline (see an example in Fig. 6). In the Southern Hemisphere, these extremely heavily calcified morphotypes were only previously observed at the Pacific border of southern Patagonia (in the Archipelago Madre de Dios Fjord area) and in the Northern Hemisphere, in Norwegian fjords (e.g. Young et al., 2014). Diaz-Rosas et al. (2021) suggested that the R/hyper-calcified morphotype has a marginal ecological niche preference compared to moderately calcified types A and A-CC. Therefore, the few specimens of *E. huxleyi* type A overcalcified (i.e. heavily calcified looking in between the R/hyper-calcified and/or A-CC morphotypes by Diaz-Rosas et al. (2021)) observed in this work, and by Saavedra-Pellitero et al. (2019) in the Drake Passage, could be attributed to different niches overlapping offshore of Chile.

The normal type A specimens show a moderate range of variation in tube width, comparable to type O, but smaller than B, B/C-C, with type C having the thinnest tube width. The distal shield element width and the number of T-elements of the different specimens are closely related to the length and width measured (Fig. 6) as they are all indicators of coccolith size. There is broader variation in coccolith size (length and width) within morphogroup B compared to morphogroup A, which is more restricted. Suchéras-Marx et al (2022) pointed out that *E. huxleyi* coccolith size is limited by the cell diameter because heterococcoliths are produced intracellularly and are extruded later on. Interestingly, specimens of *E. huxleyi* type A in the New Zealand transect are notably smaller than those offshore of Chile, which we link to local adaptations, seasonality and even ecological interactions such as predation (e.g. Monteiro et al., 2016; Hansen et al., 1996).

However, the coccolithophore assemblages in the PFZ and south of it are monospecific, which is also known from other areas of the SO (e.g. Charalampopoulou et al., 2016), and consist almost entirely of *E. huxleyi* morphogroup B. The mean placolith length of *E. huxleyi* morphogroup B (including types B, B/C-C, and O) in both transects is very similar (Drake Passage: 2.98 ± 0.40 μm, New Zealand: 2.87 ± 0.35 μm) and agrees well with the corresponding B/C measurements of Charalampopoulou et al. (2016) in samples retrieved in 2009 in the Drake Passage (2.8 μm). Still, our averaged values are slightly lower than the mean length estimated by Poulton et al. (2011) on the Patagonian Shelf (3.25 ± 0.40 μm). This could be due to the fact that Poulton et al. (2011) did not distinguish between types B and O (i.e. they were merged into B/C), which are typically larger coccoliths than B/C (Fig. 6) and could have contributed to increase the averaged length. The length range for types B/C-C (Drake Passage: 2.20 to 3.98 ± 0.37 μm, New Zealand: 1.95 to 3.62 ± 0.33 μm) agrees quite well with the range reported by Cook et al. (2011) for cultured B/C strains (2.65 to 4.80 μm) and is in the range of sizes presented by Charalampopoulou et al. (2016) for the Drake Passage (1.8 to 5.5 μm). The fact that we record lower values is simply a matter of taxonomical considerations regarding the overlapping sizes of morphotypes B, B/C, C and O, visually represented in Figure 6.

487

Overall, our morphometric data from selected samples along the New Zealand and Drake Passage transects show (1) differences in calcification between the different *E. huxleyi* morphotypes, which are particularly evident in type A (Figs. 3, 4, 5, and 6), (2) a large scatter of relative tube width within morphotypes and within each sample, particularly pronounced in the New Zealand transect (Figs. 4, 6), and (3) a slight decreasing trend in coccolith size and degree of calcification in the Drake Passage (Figs. 3, 6), which is not observed in the New Zealand transect. This suggests that environmental influences have no significant effect on the degree of calcification, but clearly control the distribution of *E. huxleyi* morphotypes (which are genetically-determined; Bendif et al., 2023) and thus indirectly affect the coccolith mass variation. This could also explain the southwards decreasing trend in calcification in the Drake Passage, as the relatively large and heavily calcified type A coccospheres occur in the northern SAZ

The different taxonomic considerations of *E. huxleyi* in different studies make it difficult to compare and combine data, especially in light of recent advances in the field. Given the dominance of this taxa in the SO, a key area for global warming and ocean acidification studies, efforts by the scientific nannofloral community should focus on a more uniform classification of *E. huxleyi* morphotypes. However, differentiation and recognition of the various morphotypes is time consuming and tedious and plays only a minor role in the calculation of the total coccolithophorid PIC, as observed in other areas of the SO (e.g. Rigual Hernández et al., 2020a, b). The changes in mass within the B morphotype (with types B/C-C, C, O) in the two transects are negligible in the PIC calculation, while a differentiation into morphogroups A and B has an influence on the calculation of the PIC. However, specimens of *E. huxleyi* belonging to morphogroup A only occur in the northern areas of both transects, where they play a role together with the PIC input from other massive species such as *C. leptoporus* (Fig. 5). Overall, the changes in total coccolithophore-PIC in the study area are caused by the abundance and occurrence within the entire coccolithophore community. The relative contribution of the different *E. huxleyi* morphogroups to the coccolithophore-PIC in the SO deserves further exploration in light of the rapid development of remote sensing and recent evolution of machine learning approaches for PIC estimates.

**6 Conclusions**

The comparison between particulate inorganic carbon (PIC) derived from satellite data and in situ coccolithophore-based estimates in two transects of the Pacific sector (separated in time and space) demonstrates the limited availability of high-quality satellite-derived data (mostly due to atmospheric conditions), and the need for refining methodologies to accurately produce coccolith-estimated PIC. Based on our data the following conclusions can be drawn:

1) We found that satellite-derived PIC values and coccolith-estimated PIC values follow a comparable trend in the Subantarctic Zone (SAZ) and Polar Front Zone (PFZ). However, satellite-derived PIC values are generally higher

than coccolith-estimated PIC. This difference could be due to a lack in precision in the coccolith-based PIC estimates, to the presence of foraminifera and/or other hard-shelled micro-zooplankton adding potential error when calculating total PIC volume, or to a certain bias in the algorithm due to the low number of measurements used for the validation of the satellite-derived PIC calibration (all of which were taken in the Atlantic Ocean).

2) There is an observed decoupling of satellite-derived PIC and coccolith-estimated PIC south of the Polar Front (PF), in the Antarctic Zone (AZ). Despite having satellite high reflectance values, no coccolithophores were observed in this area of high chlorophyll a concentration. We are unable to determine the reason for this with our data, but note that an abundance of small biogenic opal particles, such as small-size diatoms, silicoflagellates and/or siliceous plankton (as well as their fragments) or, potential biogenic particles not visible in scanning electron microscope (e.g. *Phaeocystis* aggregations, microbubbles, etc.) could possibly provide an explanation for this observation.

3) *Emiliania huxleyi* is the predominant coccolithophore species contributing the most to the total sea-surface coccolith-PIC in the New Zealand transect (mainly sampled in 2005) and as well as in the Drake Passage (sampled in 2016). *Calcidiscus leptoporus* may occasionally contribute significantly to the total coccolithophore-PIC at certain locations, whereas the rest of the coccolithophore taxa contribute only marginally in the studied areas.

4) *Emiliania huxleyi* consists of several morphotypes, which have different, partly overlapping geographical distributions. The relatively massive type A occurs in the northern SAZ and occasionally in the PFZ of the Drake Passage, while specimens of the less calcified morphogroup B (which includes types B, B/C, C and O) occur in the SAZ and the PFZ of both transects, but disappear drastically south of the PF. But neither the slightly different carbonate masses nor the southward changes in morphotype composition have a decisive influence on the coccolith-estimated PIC, which is only determined by the abundance of *E. huxleyi* in this area.

The satellite-derived and coccolith-estimated PIC discrepancies observed in this work emphasize the importance of in situ measurements and sampling; it also highlights the need for further investigation to fully understand the factors influencing water-leaving radiance and the reliability of remote sensing estimates, especially south of the PF. Future research should focus on refining methodologies and satellite algorithms to improve the accuracy of PIC estimates and better understand the dynamics of coccolithophores as well generally phytoplankton and zooplankton communities in the Pacific sector of the Southern Ocean (especially compared to other sectors). Such efforts will enhance our understanding of carbon cycling and its impact on marine ecosystems at high latitudes.

**Acknowledgements**

Stellite-derived Particulate Inorganic Carbon (PIC) data was downloaded from the Ocean Color Web Level 1 & 2 Browser (https://oceancolor.gsfc.nasa.gov/cgi/browse.pl?sen) and Level 3 & 4 Browser (https://oceancolor.gsfc.nasa.gov/l3), both of them services provided by NASA's Ocean Biology Distributed Active Archive Centre (OB.DAAC).

The authors acknowledge the use of the JASMIN (Joint Analysis System for the Met Office, NERC, and UKRI) (https://jasmin.ac.uk/) Jupyter Notebook service to process the satellite-derived PIC data. We would like to express our gratitude to the JASMIN team for their support and the valuable resources they provide to the scientific community.

The authors are grateful to two anonymous reviewers and to the handling associate editor, Prof. Shutler, for their invaluable suggestions on a previous version of the paper. The Alfred Wegener Institute Bremerhaven provided part of the plankton samples required for this study. Frank Lamy, Hartmut Schulz, *R/V Polarstern* officers and crew are thanked for their help during the PS97 Expedition.

Dr. Frigola (Barcelona Supercomputing Center, Spain), Dr. Merkel (University of Bremen/MARUM, Germany) and Dr. Hardiman (University of Portsmouth, UK) are acknowledged for their help with remote sensing data collection advice. Dr. Pepin (University of Portsmouth, UK) and Dr. Balch (Bigelow Laboratory for Ocean Sciences, USA) are thanked for their comments and suggestions on this piece of research during the "Advances in Coccolithophore research" meeting. Dr. Saavedra (RIP) is thanked for his continuous encouragement to finish up this paper.

**Financial support**

This research was supported by the University of Portsmouth, by the Deutsche Forschungsgemeinschaft with a grant to Karl-Heinz Baumann (reference number: BA 1648/30-1) -through previous funding for Mariem Saavedra-Pellitero and Nele M. Vollmar- and by the MIUR project "Dipartimenti di Eccellenza 2018/2023" for Elisa Malinverno, at the Department of Earth and Environmental Sciences, University of Milano-Bicocca. The University of Portsmouth Research and Innovation Services, as well as Copernicus Publications are acknowledged for their additional financial support to publish this paper as Open Access.

**Data Availability Statement**

The authors confirm that the data from which the findings of this study are available within the article Supplementary Materials and are stored in the data repository https://pangaea.de/ (https://doi.pangaea.de/10.1594/PANGAEA.964672 and https://doi.pangaea.de/10.1594/PANGAEA.964674)

**Author contributions**

The study was designed by EM, MSP and KHB. EM and NMV carried out the morphometric measurements and classified the specimens of *E. huxleyi.* EM and MSP calculated coccolith-PICs, plotted the data and wrote an earlier version of the

manuscript. NB-J and HL provided remote sensing data for the study area, and were actively involved in the discussion of the findings as well as in the writing of the paper. All authors approved the submitted version.

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

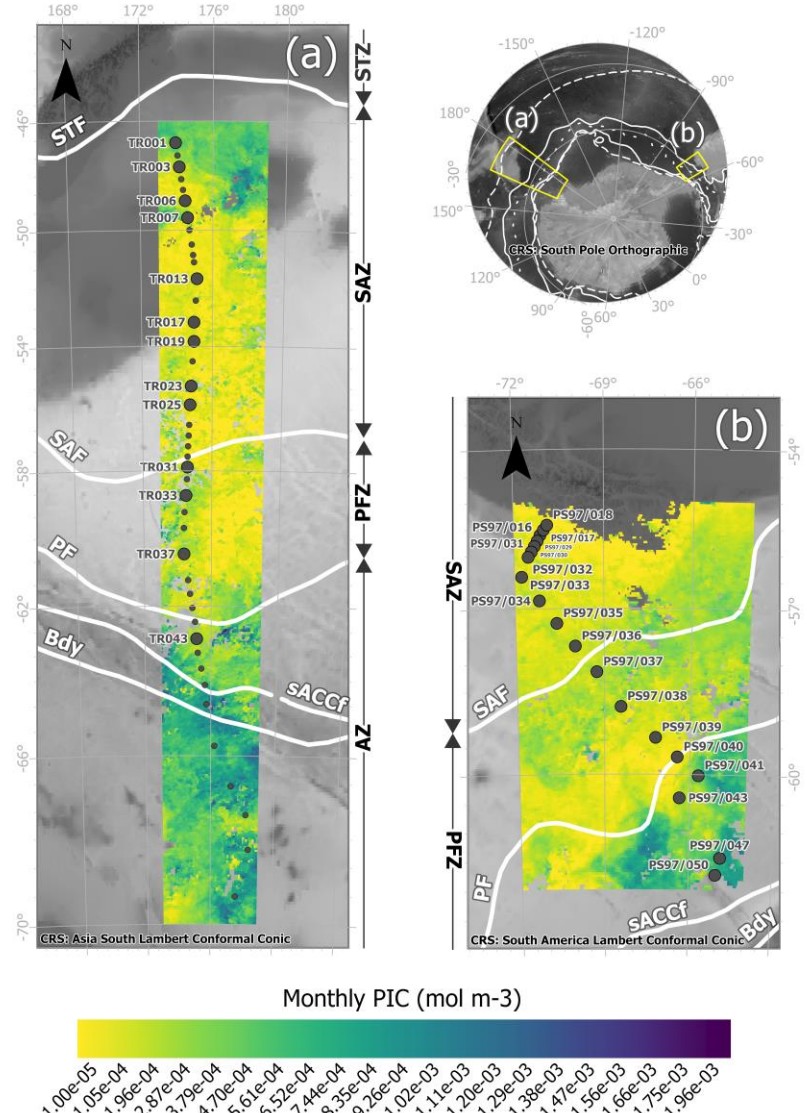

Monthly PIC (mol m-3)

Figure 1: Study area showing the location of the water samples retrieved from (a) the New Zealand transect, collected during the XX Italian Expedition from New Zealand to Antarctica on board *R/V Italica* (December 2004-January 2005) and (b) the Drake Passage transect, collected during *Polarstern* Expedition PS97 across the Drake Passage (February-March 2016). Large dots indicate samples in which biometries on *Emiliania huxleyi* were performed, and small dots where coccolithophore census were available. The maps show MODIS-Aqua L3 PIC concentrations in mol m$^{-3}$ g corresponding to (a) monthly mean over January 2005 and (b) monthly mean over February and March 2016, overlain on a bathymetry background (GEBCO Compilation Group, 2022). White lines indicate the average position of the ACC fronts (Orsi and Harris, 2019), from north to south these are: SAF (Subantarctic Front), PF (Polar Front), sACCf (Southern ACC Front) and Bdy (Southern Boundary). The Southern Ocean zones are labeled on the side of each map: STZ, Subtropical Zone; SAZ, Subantarctic Zone; PFZ, Polar Frontal Zone; AZ, Antarctic Zone.

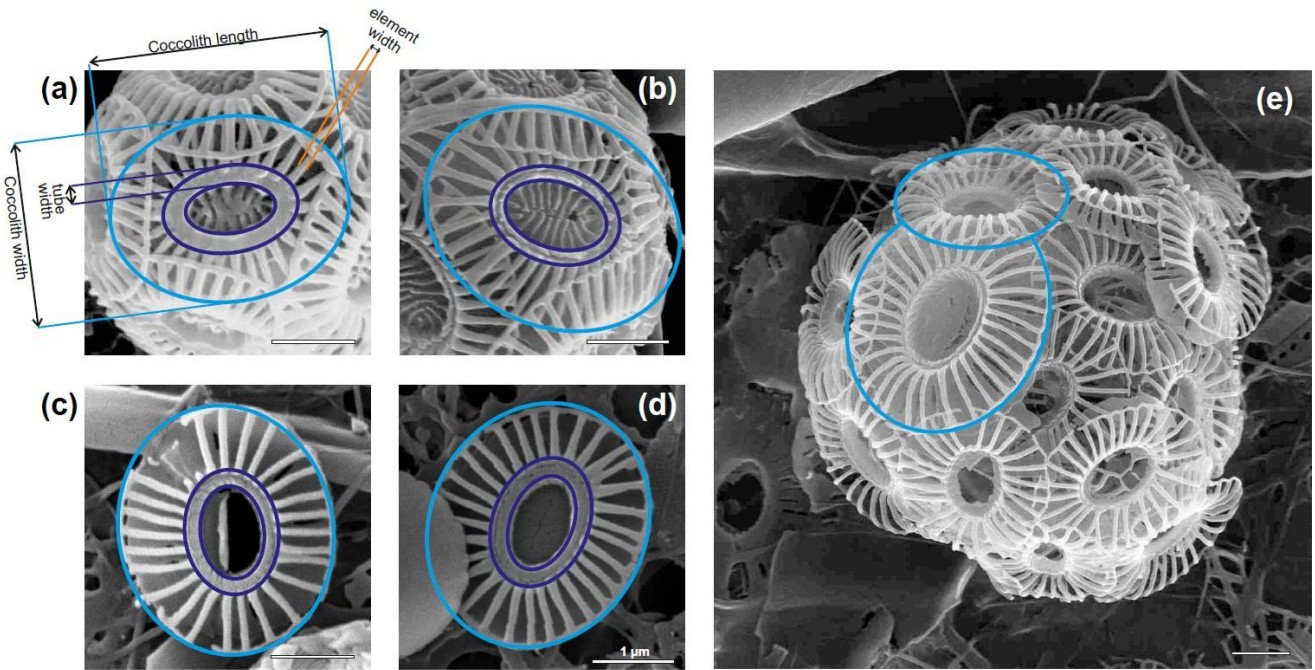

Figure 2: Parameters measured in *Emiliania huxleyi* coccoliths (a, b) type A and (c, d, e) type O in plankton samples from the New Zealand transect. Note the coccolith size variation in (e) within the same coccosphere.

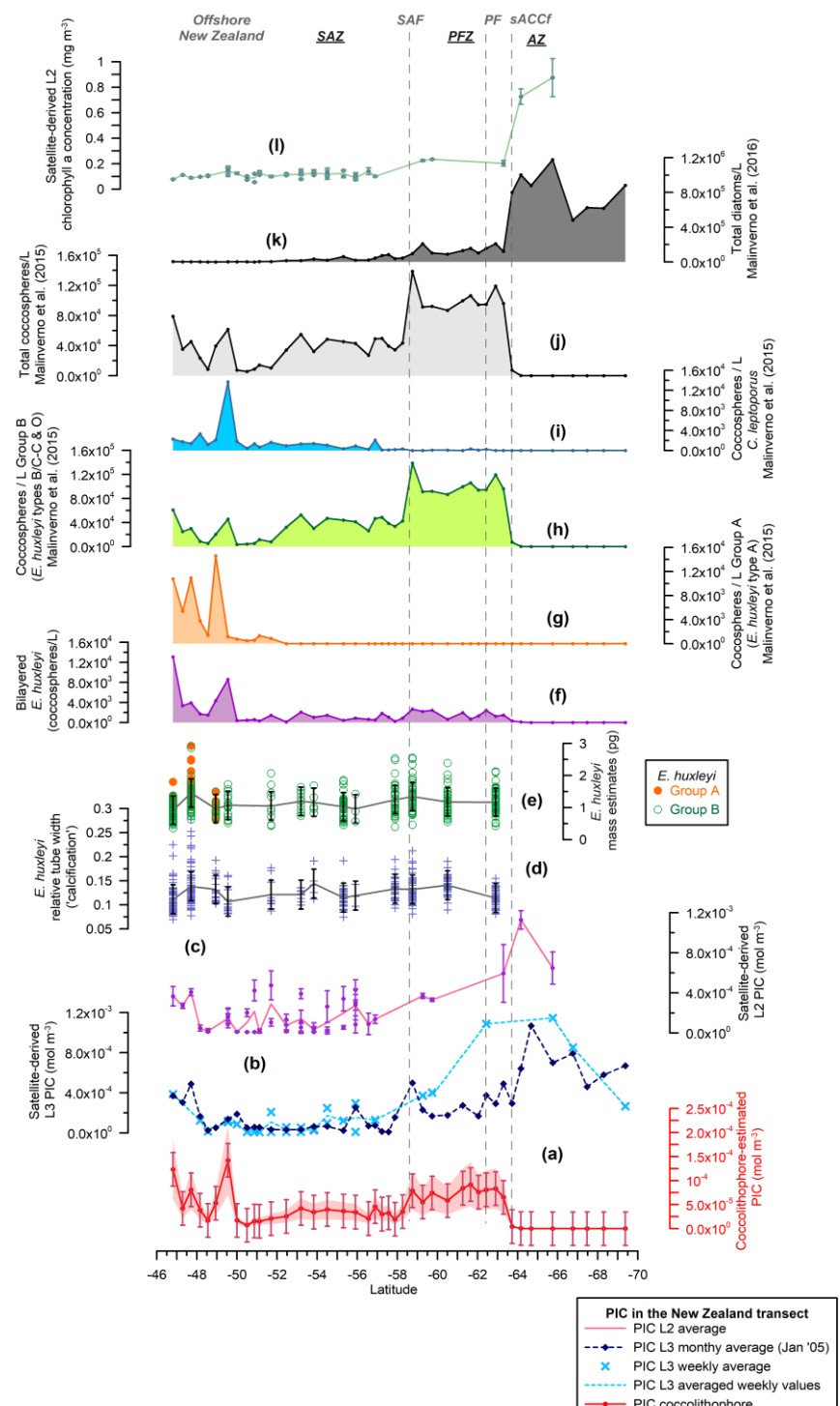

**Figure 3: New Zealand transect showing (a)** estimated total coccolithophore PIC (red line with dots) in mol m⁻³, **(b)** MODIS-
Aqua L3 PIC concentration values (mol m⁻³) corresponding to a monthly average (January 2005, dark blue dashed line

with diamonds), weekly average (light blue dashed line with crosses), (c) MODIS-Aqua L2 PIC concentration values in mol m$^{-3}$ (average in pink) (d) *Emiliania huxleyi* relative tube width index (average in gray), (e) *E. huxleyi* coccolith mass estimates (pg) for morphogroup A (dots) and B (circles) (average in gray), (f) number of bilayered *E. huxleyi* (coccospheres/L), (g) number of *E. huxleyi* morphogroup A (coccospheres/L), (h) number of *E. huxleyi* morphogroup B (coccospheres/L), (i) number of *Calcidiscus leptoporus* (coccospheres/L), (j) Number of total coccolithophores (coccospheres/L) (Malinverno et al., 2015), (k) Number of total diatoms (cells/L) (Malinverno et al., 2016), (l) MODIS-Aqua L2 chlorophyll a concentration in mg m$^{-3}$(average in light green).Note that the plankton samples were retrieved at ca. 3 m water depth. Vertical bars indicate one standard deviation on the entire population in (a), (d) and (e), and the standard deviation (considering a 5 x 5 window) in (c) and (l). The shaded area in (a) represents a 50% error. Vertical dashed lines indicate some of the ACC fronts (Orsi and Harris, 2019): SAF (Subantarctic Front), PF (Polar Front) and sACCf (Southern ACC Front). The Southern Ocean zones are labeled as SAZ (Subantarctic Zone), PFZ (Polar Frontal Zone) and AZ (Antarctic Zone).

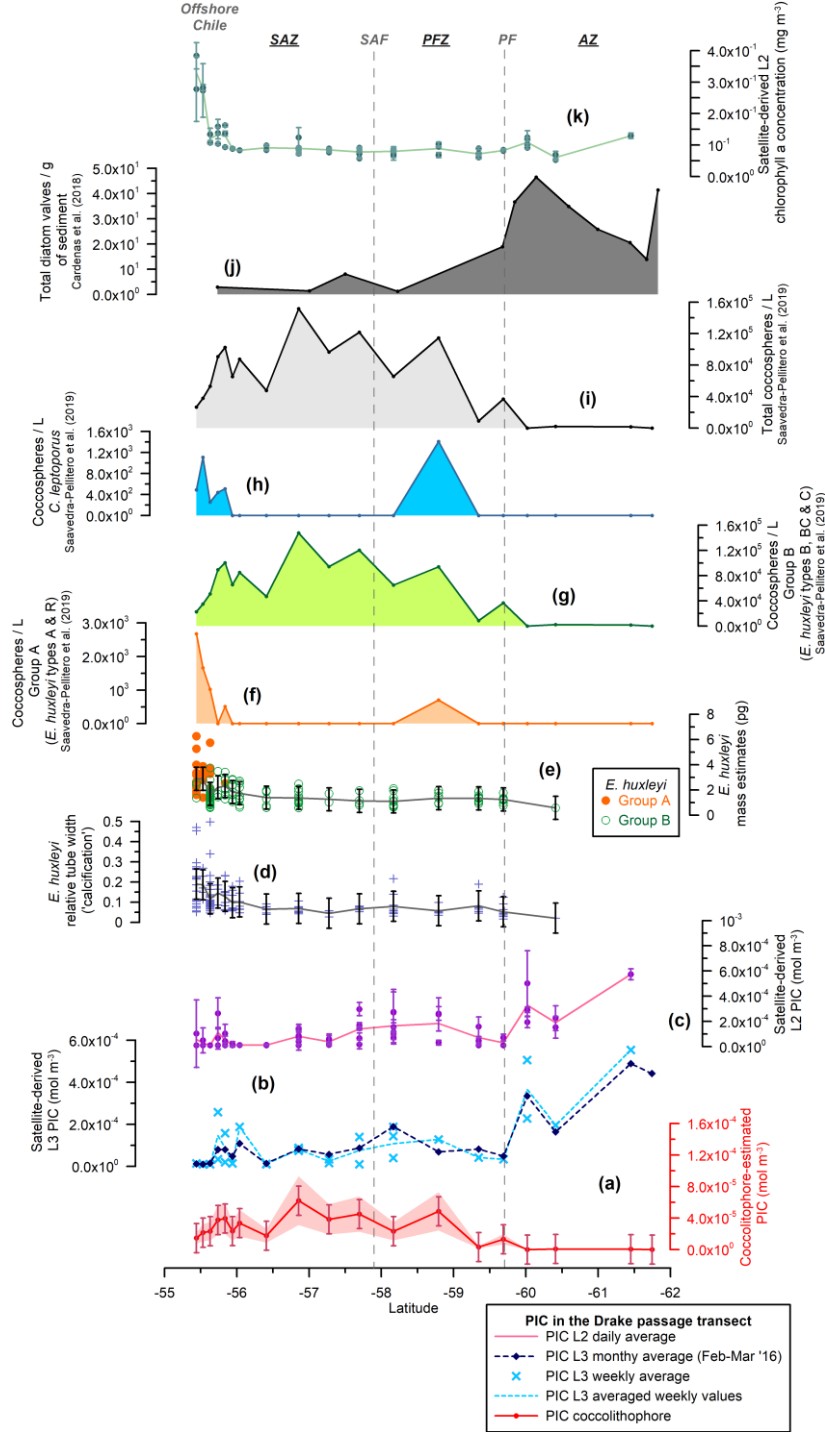

**Figure 4: Drake Passage transect showing (a) estimated total coccolithophore PIC (red line with dots) in mol m⁻³, (b) MODIS-Aqua L3 PIC concentration (mol m⁻³) corresponding to a monthly average (February and March 2016, dark blue dashed line with diamonds), weekly average (light blue dashed line with crosses), (c) MODIS-Aqua L2 PIC concentration in mol m⁻³ (average in**

pink), (d) *Emiliania huxleyi* relative tube width index (average in gray), (e) *E. huxleyi* coccolith mass estimates (pg) for morphogroup
A (dots) and B (circles) in (average in gray), (f) number of *E. huxleyi* morphogroup A (coccospheres/L), (g) number of *E. huxleyi*
morphogroup B (coccospheres/L), (h) number of *Calcidiscus leptoporus* (coccospheres/L), (i) Number of total coccolithophores
(coccospheres/L) (Saavedra-Pellitero et al., 2019), (j) Number of valves per gram of sediment from surface sediment samples across
the Drake Passage and Scotia Sea (Cárdenas et al., 2018), (k) MODIS-Aqua L2 chlorophyll a concentration in mg m$^{-3}$ (average in
light green). Note that plankton samples were retrieved at 5, 10 and 20 m water depth. Vertical bars indicate one standard deviation
on the entire population in (a), (d) and (e), and the standard deviation (considering a 5 x 5 window) in (c) and (k). The shaded area
in (a) represents a 50% error. Vertical dashed lines indicate some of the ACC fronts (Orsi and Harris, 2019): SAF (Subantarctic
Front) and PF (Polar Front). The Southern Ocean zones are labeled as SAZ (Subantarctic Zone), PFZ (Polar Frontal Zone) and AZ
(Antarctic Zone).

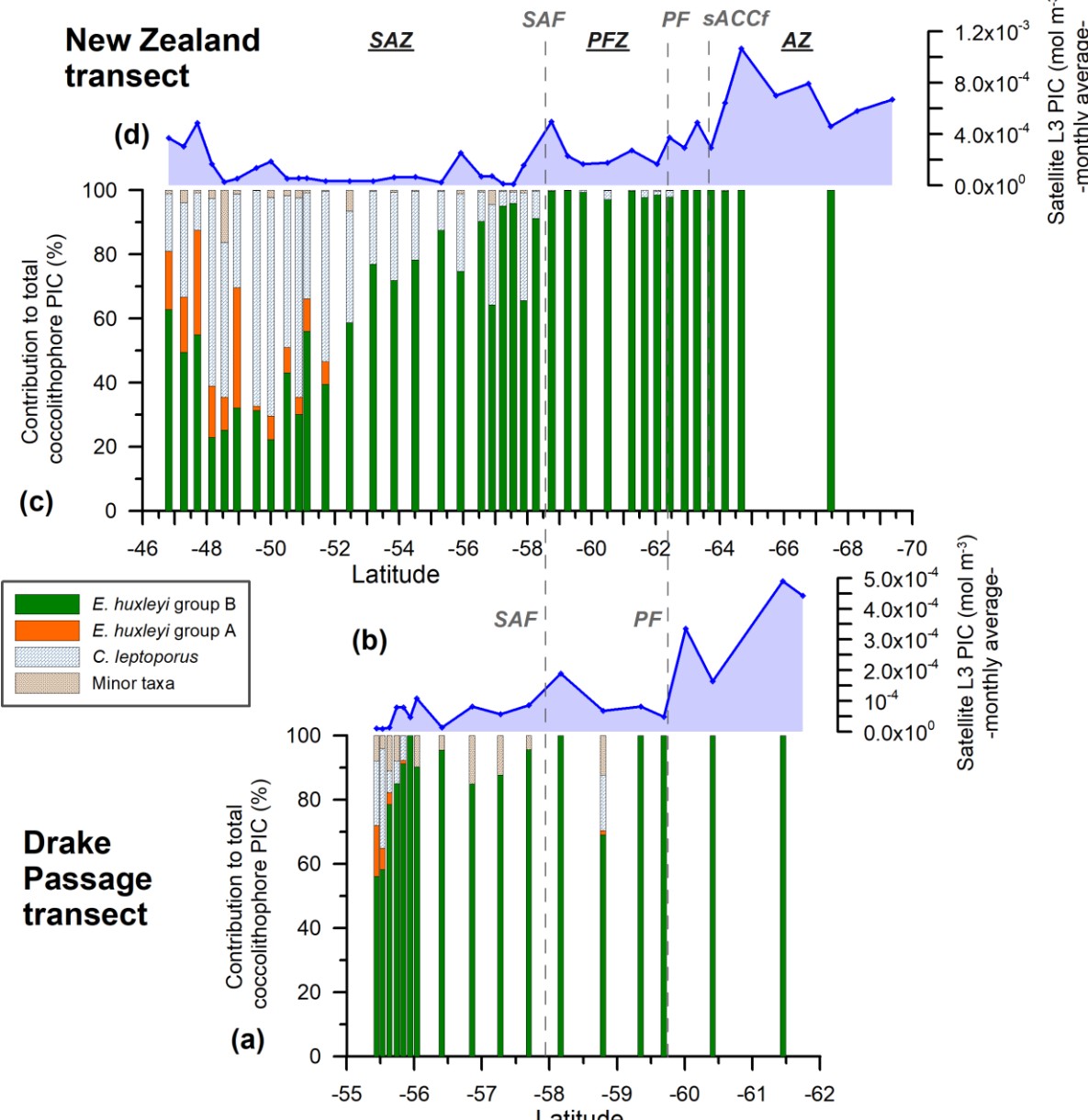

**Figure 5: New Zealand (NZ) and Drake Passage (DP) transects showing (a, c) the relative PIC contribution of the different nannofloral taxa (*E. huxleyi* morphogroups A and B, *Calcidiscus leptoporus* and minor species) to the estimated coccolithophore PIC in 38 NZ and 17 DP samples bearing coccospheres; (b, d) MODIS-Aqua L3 monthly average satellite-derived PIC values (February and March 2016, dark blue line with diamonds) in mol m$^{-3}$. Vertical dashed lines indicate some of the ACC fronts (Orsi and Harris, 2019): SAF (Subantarctic Front) and PF (Polar Front). The Southern Ocean zones are labeled as SAZ (Subantarctic Zone), PFZ (Polar Frontal Zone) and AZ (Antarctic Zone).**

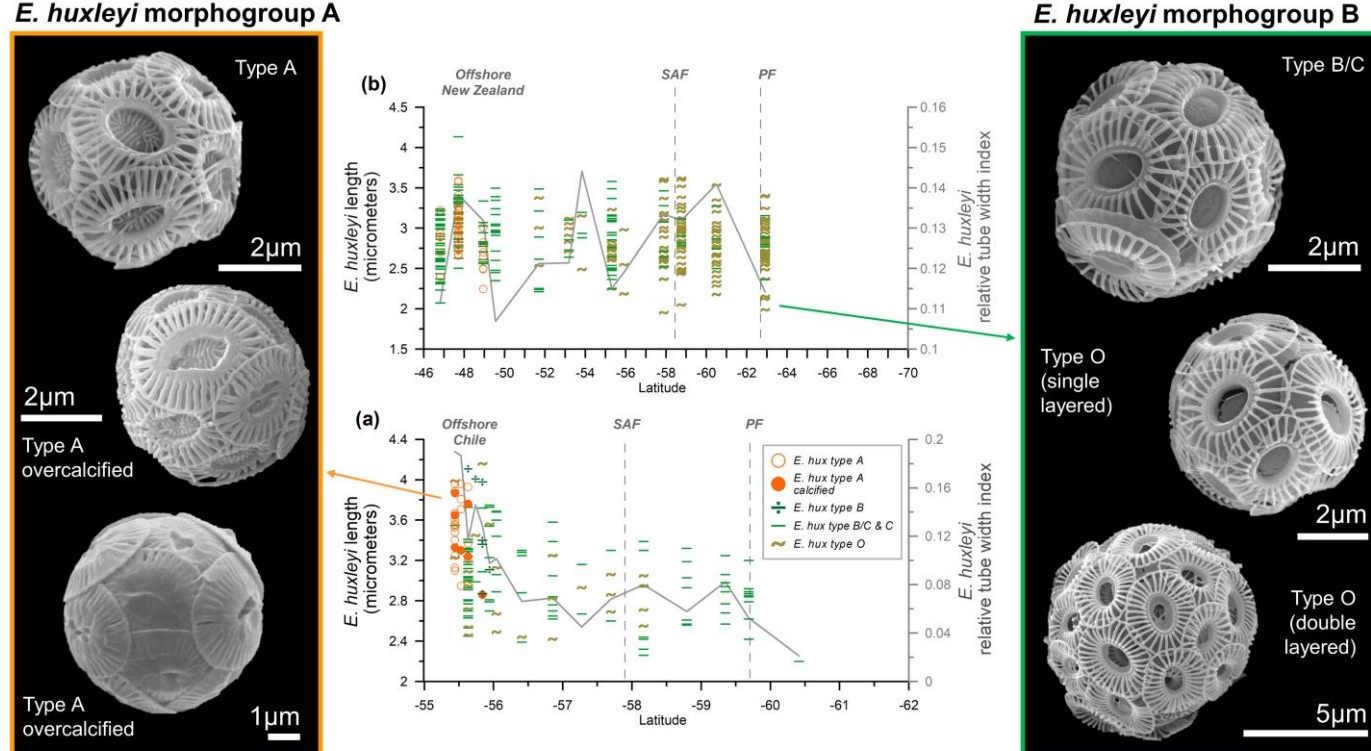

Figure 6: *Emiliania huxleyi* length (in μm) (indicated with different symbols depending on the type, and different colors depending on the morphogroup _A or B-) and averaged relative tube width index (gray line) in (a) the Drake Passage and (b) New Zealand transects. On the left-hand side: pictures of coccospheres of *E. huxleyi* type A (within the morphogroup A) showing different degrees of calcification and on the right-hand side pictures of type B/C as well as type O belonging to the morphogroup B. All the images of coccospheres are from the New Zealand transect, except for the left bottom one, which was retrieved offshore of Chile.


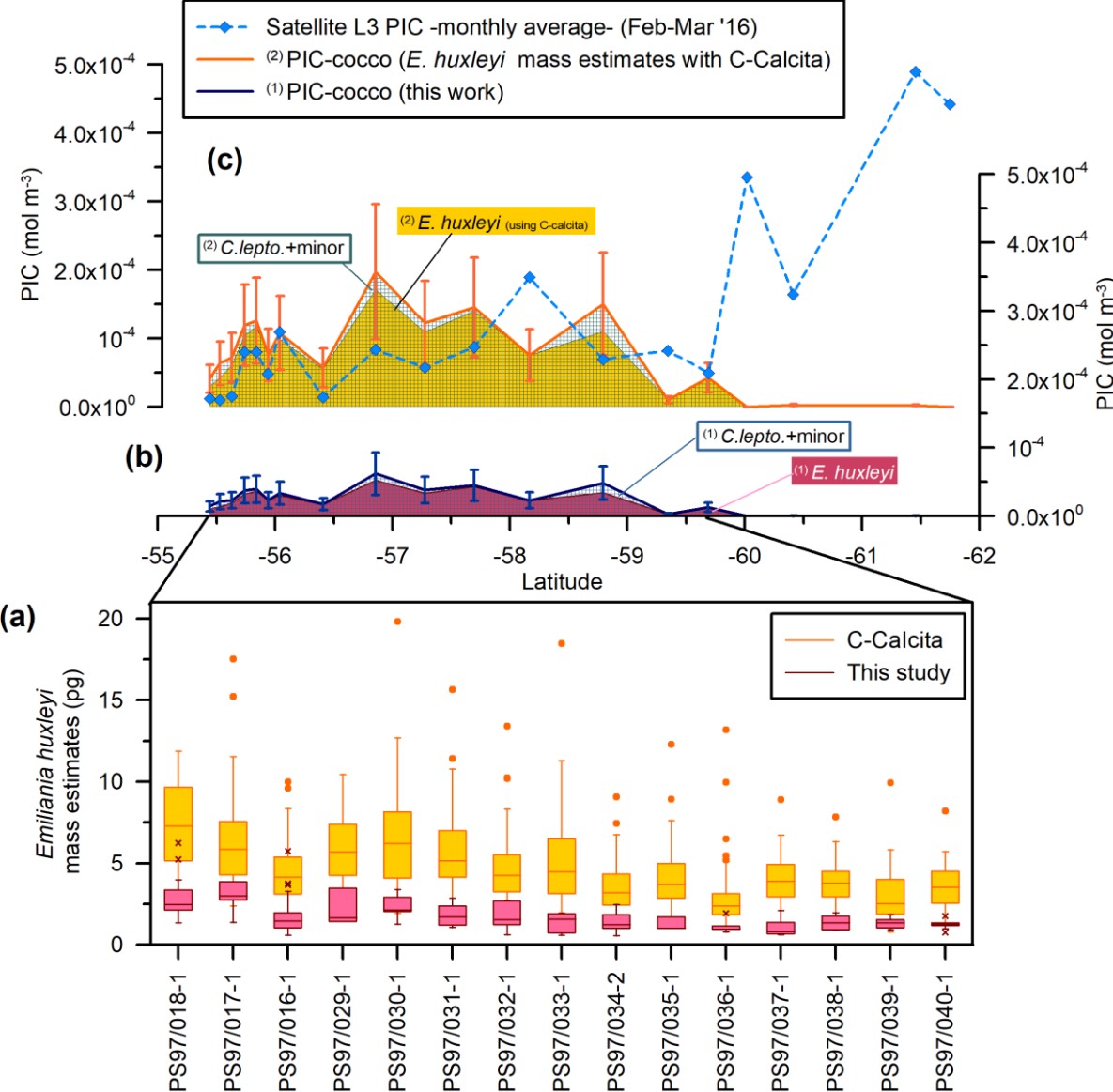

**Figure 7: Drake Passage latitudinal transect showing (a) coccolith mass estimates box plots (in pg): in dark red plus pink for this**
**study (outliers are indicated with "x") and yellow plus orange for Saavedra-Pellitero et al. (2019) (outliers are indicated with a dot);**
**(b) estimated coccolithophore PIC (PIC-cocco) (all in mol m⁻³) -this study [(1)]-; (c) MODIS-Aqua L3 monthly average satellite-derived**
**PIC values (blue dashed line with diamonds) and PIC-cocco calculated considering averaged *Emiliania huxleyi* mass estimates**
**obtained with the software C-Calcita [(2)] (Saavedra-Pellitero et al., 2019). Note that the contributions of different coccolith taxa or**
**groups have been indicated (*C.lepto.* = *Calcidiscus leptoporus;* minor = minor species) and that the data is stacked for each of the**
**approaches. Vertical bars in (b) and (c) represent a 50% error.**

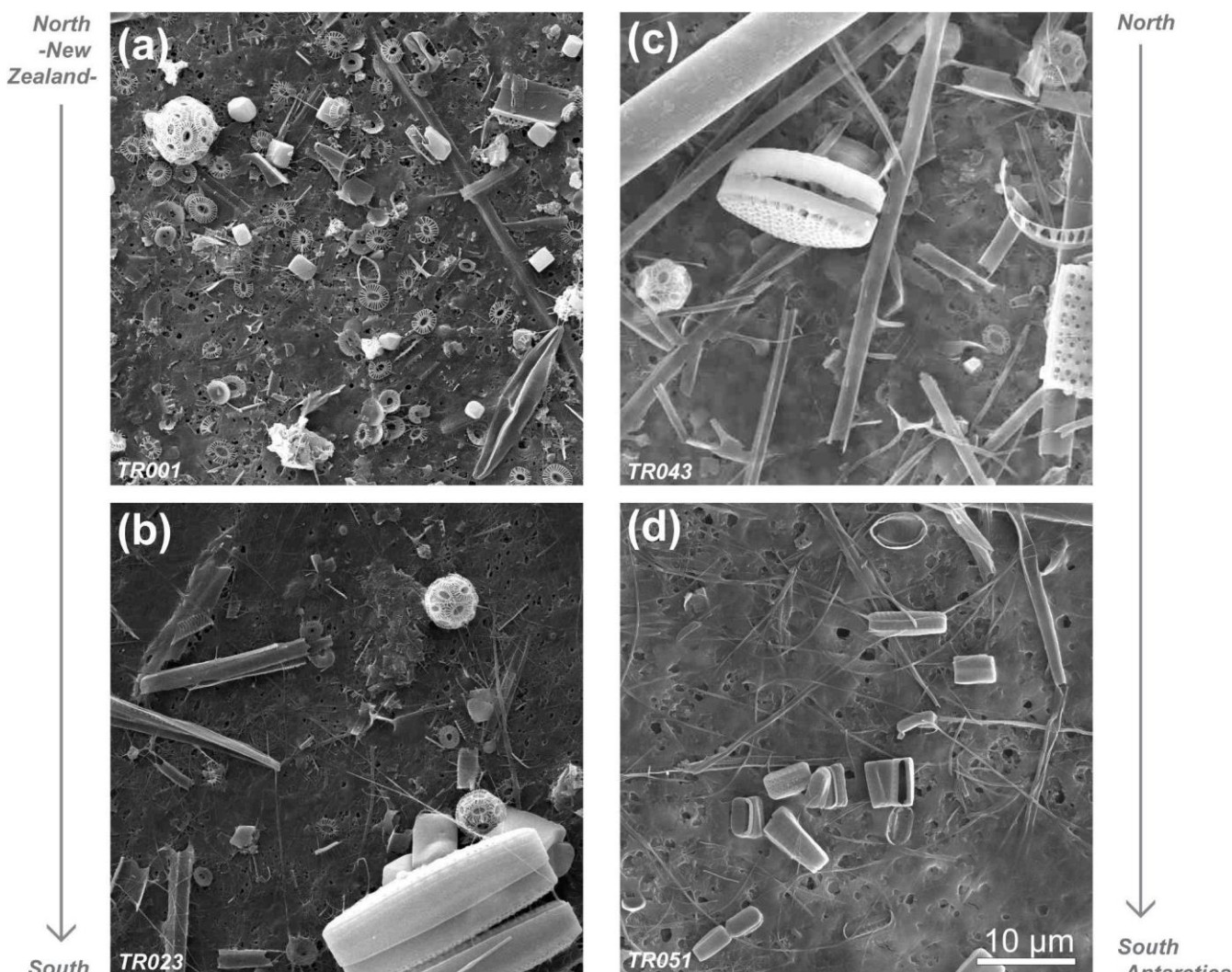

**Figure 8: SEM pictures of samples retrieved in the Subantactic Zone (a, b) and south of the Polar Front (c, d) in the New Zealand**
**transect.**


**Table 1: Overview of the samples considered for this study, including the sampling area, number of plankton samples considered**
**for this study, expedition, research vessel, water sampling dates, coordinates and data already available from previous publications.**

| Area | Number of samples considered for this work | Water depth (m) | Expedition | Research Vessel | Plankton sampling period | Coordinates | Previous publications |
|---|---|---|---|---|---|---|---|
| New Zealand | 42 | 3 | XX Italian Expedition | R/V Italica | 31.12.2004-06.01.2005 | 46.81°S to 69.98°S | Coccolithophore assemblages: Malinverno et al. (2015); Dinoflagellates, Coccolithophores, Silicoflagellates, Diatoms, Parmales, Archaeomonads and micro-zooplankton: Malinverno et al. (2016) |
| Drake Passage | 19 | 5, 10 and 20 | Expedition PS97 | Polarstern | 24.02.2016-05.03.2016 | 55.44°S to 61.75°S | Coccolithophore assemblages: Saavedra-Pellitero et al. (2019); *Emiliania huxleyi* mass estimantes: Saavedra-Pellitero et al. (2019) |


**Table 2: Length, shape factors (Ks) and number of coccoliths per coccosphere used in this work for the New Zealand transect and the Drake Passage transect. (*) Indicates an average of the number of coccoliths per coccosphere. Note that the different Ks used here were mostly based on Young and Ziveri (2000). The shape factor for morphotype O (Ks = 0.015) was introduced by Poulton et al. (2011) in a plankton study along the Patagonian Shelf for a morphotype with a central area described as an "open or thin plate" which the authors called type B/C but that we identified as morphotype O.**

| Coccolithophore species | Average length ± standard deviation (μm) New Zealand | Average length ± standard deviation (μm) Drake Passage | Source | Ks | Source | Number of coccoliths per coccoshere N. Zealand | Number of coccoliths per coccoshere Drake P. | Source |
|---|---|---|---|---|---|---|---|---|
| *Calcidiscus leptoporus* spp. *leptoporus* | 5.7±0.6 | 5.7±0.6 | This work (biometries offshore N. Zealand) | 0.08 | Young and Ziveri (2000) | 15 | 15 | Kleijne (1993) |
| *Emiliania huxleyi* group A (average value) | | | | 0.03 | This work | | | |
| *Emiliania huxleyi* A overcalcified | 2.95±0.28 | 3.49±0.33 | This work | 0.04 | Young and Ziveri (2000) | 15 single layered, 35 double layered | 25 (*) | This work (own observations) |
| *Emiliania huxleyi* A (normal) | | | | 0.02 | Young and Ziveri (2000) | | | |
| *Emiliania huxleyi* group B (average value) | | | | 0.02 | Young and Ziveri (2000) | | | |
| *Emiliania huxleyi* B-B/C-C | 2.87±0.35 | 2.98±0.40 | This work | 0.02 | Young and Ziveri (2000) | | | |
| *Emiliania huxleyi* O | | | | 0.015 | Poulton et al. (2011) | | | |
| *Gephyrocapsa muellerae* | 3.9 | 3.9 | Young and Ziveri (2000) | 0.05 | Young and Ziveri (2000) | 15 | 15 | Samtleben & Schroder (1992) |
| *Syracosphaera* spp. | 2.2 | 5.5 | Young and Ziveri (2000) | 0.03 | Young and Ziveri (2000) | 25 | 25 | Okada & McIntyre (1977) |
| **Minor taxa** | | | Young and Ziveri (2000) | | Young and Ziveri (2000) | | | Yang & Wei (2003) |

**Table 3. Summary of MODIS-Aqua products used in this study. (**)The first 8-day period of each year always begins with January 1, the second with January 9, the third with January 17, etc. The final "8-day" composite of each year comprises only five days in non-leap years (27 - 31 December) or six days in leap years (26 - 31 December) (NASA Ocean Biology Processing Group, 2018).**

| Satellite product | Biophysical variable | Extraction method | Time period | Drake passage transect | | New Zealand transect | |
|---|---|---|---|---|---|---|---|
| | | | | Time span | Num. of scenes | Time span | Num. of scenes |
| MODIS-A Level 2 | ▪ PIC concentration (mol m$^{-3}$) ▪ Chlorophyll a concentration (mg m$^{-3}$) | mean of 5x5 window centered on measurement location | Daily timestamp | 17-02-2016 / 12-03-2016 | 50 | 24-12-2004 / 13-01-2005 | 34 |
| MODIS-A Level 3 | PIC concentration (mol m$^{-3}$) | value of pixel enclosing measurement location | 8-daily(**) timestamp | 10-02-2016 / 12-03-2016 | 4 | 26-12-2004 / 08/01/2005 | 2 |
| MODIS-A Level 3 | PIC concentration (mol m$^{-3}$) | value of pixel enclosing measurement location | Monthly timestamp | 01-12-2004 / 31/01/2005 | 2 | 01-02-2016 / 31/03/2016 | 2 |