# Peer review of "Mismatch between coccolithophore-based estimates of particulate"

_EGUsphere, 2023_

## Editor Comment (EC1)

Review of "Pacific Southern Ocean coccolithophore-derived particulate inorganic carbon (PIC): A novel comparative analysis of in-situ and satellite-derived measurements".  By Saavedra-Pellitero et al.                                    Submitted to EGUsphere

Synopsis:  This paper evaluates the coccolithophores in 13 water samples from a transect south of New Zealand to Antarctica (Dec. 2004 to Jan. 2005) plus19 more water samples from a transect at the western end of Drake Passage (Feb.-March 2016).  They made SEM images of the water samples, then calculated the PIC concentration of each sample using the morphometric properties of the coccoliths.  Then they used level-3 satellite data to estimate PIC concentrations.  These satellite data were binned to 4km-resolution and a mix of temporal binning (daily, 8-day average and one-month time-averaged satellite images but mostly one-month binning).  They then concluded that Subantarctic waters and waters north of the Polar Front, the NASA PIC algorithm worked reasonably well.  South of the Polar Front, however, there were few coccolithophores yet the satellite observations still suggested elevated PIC.  The authors suspect that other reflective particles than PIC were causing the PIC algorithm to be in error.

General comments
This paper deals with an interesting topic, the coccolithophores of the Southern Ocean and what appear to be false, high-concentrations of coccolithophores as derived by satellite when the coccolithophore concentrations were observed to be low in their 32 water samples.  Unfortunately, the paper is lacking in some fundamental key aspects and this reviewer cannot recommend publication it its current form.  The key shortfalls are: (1) the NASA PIC algorithm measures the concentration of suspended PIC.  Nowhere in the paper did the authors ever actually measure PIC analytically in order to judge the errors in the satellite approach, (2) the authors "measured" PIC using an approach based on the morphology of coccoliths and the numbers of coccoliths observed on coccolithophore cells under the SEM, calculating PIC volume.  Nowhere did they use analytically-derived PIC estimates to examine the error of this SEM-based approach, (3) the remote sensing approach used here was inadequate for estimating PIC in these Southern Ocean waters, (4) their conclusion that the PIC algorithm does not perform well south of the Polar Front leads them to hypothesize that something else is causing the error. They hypothesized in the abstract that the reflectance signal from these waters was "due to the prevalence of small opal particles or unknown highly reflective particles (such as *Phaeocystis* aggregations) or suspended sediment " and later in the paper, they mention other possible sources of errors such as microbubbles, floating loose ice or high concentrations of other particulate matter such as glacial flour", which could cause errors.  Unfortunately, barely any evidence was provided for these possibilities.  I will elaborate on each of these points below.

*Analytical PIC measurements*
        The gold-standard for measuring PIC in the ocean uses analytical measurement of either particulate Calcium (using any of several methods: particulate calcium analyses using flame photometry or optical emission spectrometry, or carbonate carbon measurements using coulometry).  Any of these techniques would have been ideal to use to absolutely validate the PIC concentrations with quantifiable accuracy and precision.

*PIC volume measurements using SEM*
The technique that these authors used (Young & Ziveri, 2000), involves SEM imaging of filtered coccolithophores for the multiple morphotypes of *E. huxleyi* (the most abundant coccolithophore species in these waters).   Using this laborious technique, they quantified the following

morphotypes for coccolithophores: Type A, Type A over-calcified, Type B, Type B/C, Type C, Type O single layered and Type O double-layered) as well as a few other coccolithophore species (Young & Ziveri, 2000). These measurements involve (for hundreds of cells) quantifying the length of the major and minor axes of the coccoliths, number of t-elements, ray width, tube width, and a shape factor. Each of these measurements will have an error, which would compound to the overall error of the coccolith volume, hence the PIC per cell. No such error bars were ever provided for the PIC measurements of coccolith morphometrics from their 32 samples. Moreover, it was critical to provide the accuracy and precision of the PIC morphometric measurements, measured against calibrated, analytical PIC measurements. As for the influence of diatoms in the Southern waters causing error in the satellite measurements of PIC, the only evidence was shown is four SEM images that contained diatoms. They never presented the abundance or biomass data of diatoms present in the samples, (nor an estimate of the biogenic silica associated with them, for that matter). It would have been nice to know some of the other features of these waters, too, such as chlorophyll concentrations, POC concentrations, TSS (total suspended sediments) to tease out what the mystery particles were causing the discrepancy.

*Remote sensing approach*
        The authors used questionable remote sensing techniques to derive the PIC concentration. It is not clear exactly what they did here and more information is needed to assess this. They did say that their satellite PIC estimates were based on Level 3 data, highly binned satellite data in space and time. For example, in creating Level 3 data, daily, one kilometer resolution measurements would have been aggregated and averaged into 4km pixels. In addition, using data at 4km resolution could likely have included multiple overpasses over a day, especially given the high latitude. Multiple overpasses mean the radiances are associated with multiple azimuthal and nadir-viewing geometries. Why is this a problem? For algorithm development, one should always use Level 2 data. These data are recovered from a single overpass, first calculated from unprocessed instrument data (Level 0), to Level 1A data (reconstructed, unprocessed instrument counts), converted to calibrated water-leaving radiance data (Level 1B) at a given solar zenith and nadir viewing angle and those radiances are finally processed to geophysical variables as Level 2 data (e.g., PIC concentrations using the PIC algorithms at hand). Using already-binned, level 3 data can lead to spurious results for relating radiances to PIC due to the already-mentioned bin-averaged data, likely collected from multiple overpasses, and potentially applying an arithmetic average when the variables are actually log normal in distribution (requiring a geometric average) or vice versa. Algorithm validation should always be done with Level 2 data as per Bailey and Werdell (2006). There's too much binning that goes on to get from L2 to L3 to do an accurate comparison. It looks like, due to abundant cloudy skies in the Southern Ocean, these authors had to use monthly composites for the most part, to get the satellite PIC data to match with their ship samples. Bailey and Werdell (2006) recommend overpasses within ±3 hours of field observations (not even ±24 hours), in order to properly validate any ocean colour-derived variable. Here, the monthly level-3 composites would have been good to a temporal resolution of ±15d for the most part, hardly a "coincident" matchup! This is a long temporal difference to validate against. One can go from a phytoplankton bloom to background phytoplankton concentrations in just a few days, especially when grazers and viruses are devouring them! An error propagation calculation is essential, for all the measurements described in this paper (satellite, PIC volume), in order to assess the true errors.

*Causes for PIC error in the satellite approach south of the Polar Front*
On both transects, the authors saw a drop-off in the PIC volume calculations south of the Polar Front. However, the possible reasons are numerous, such as dense diatom populations, dense

*Phaeocystis* aggregations, floating ice crystals, bubbles or other suspended sediments, to name a few.  Yes, they saw some diatoms in these waters south of the Polar Front (see their Fig. 9c and d) but upon close inspection, there were diatoms in the waters from north of the Polar Front, too (see their Fig. 9a and 9b).   They also saw coccolithophores in their SEM from south of the Polar Front (see Fig. 9c).  Nonetheless, they ultimately concluded that diatoms must be responsible for this error. Note, the SEM technique would never have visualized soft-bodied *Phaeocystis* in an identifiable form, ice crystals or bubbles, even if they were present. Suspended lithogenic sediments would have been visualized under SEM, however.  Finally, foramifera can be abundant PIC sources in these Southern Ocean waters, too, contributing ~50% of the PIC in waters at 55-60°S  (Trull *et al.*, 2018) but they aptly point out that due to the optical properties of large PIC particles (they don't scatter light much per unit PIC), the satellite might not see their reflectance.  But this still could be a source of error in their PIC volume calculation (which included only coccolithophores, not foraminifera or fragments of foraminfera). This paper desperately needs an error analysis so that the authors can put error bars on the PIC estimates that appear in the figures and tables.

Specific comments:
P1 L23: Spelling "Coccolith"
P1 L26: Which satellite product was used?  What sensor?
P2 L47: They should also cite the earlier references who were really the first ones to show this (Holligan *et al.*, 1993; Robertson *et al.*, 1994).
P3 L89: change word "concentrations" to "measurements".  Next sentence, change "Most" to "many" since there are a significant number of subpolar studies that enumerated free coccoliths.
P3 L94: See also (Young *et al.*, 2014)
P4 L99: The Oliver et al paper is now published (Oliver *et al.*, 2023) and it included non-bloom areas, too, I believe.
P4 L104: Does aim #1 include both plated coccolithophores as well as detached coccoliths?
P4 L117 and occurrences henceforth in the manuscript: In the field of physical oceanography, salinity data should not be reported with units of ‰.
P5 L135: Cite also (Holligan *et al.*, 2010)
P5 L143: R/V Italica should be italicized.
P5 L144: Is XX referring to the roman numeral or is this some sort of place-holder that needs to be filled in?
P5 L147-154:  The authors are critical of how few stations are available in NASA's remote sensing validation of the PIC algorithm (n=42; noting that these are clear-sky matchups, using L2 satellite data within +/- 3 hours of an overpass (see general comments above).  It should be noted, too, there is very strong under-sampling in this study, with a total of 32 water samples from which the paper is based, and these samples are based on Level 3, binned data which should not be used in satellite validation.
P6 L168: There are some details the authors might elaborate on.  A SEM only views one side of a coccosphere.  What about the coccoliths one can't see?  Is this taken into consideration with the shape factors? It looks like they approximate the coccolith volume in equation 1 as a cube ($L^3$), why not a sphere?
P6 L186: For those not familiar with the Young and Ziveri method, could you say what "tube" you are talking about?
P7 L192: What fraction of your samples used daily, 8-daily or monthly PIC values?  Was there a difference on how well the validation agreed when you used the shorter time scale validations? You should state this.  I would suspect that since you show monthly images in your Figure 1, then most of your comparisons were done with monthly images.  Is this correct?  Your comment at line 204-205 would suggest that the number of daily match-ups was extremely low.  Your table 2 lists these out but for the actual validation, if you had a daily, an 8-d and a monthly

measurement, which did you use?  Please explicitly state this!  Using 8-d composites or monthly composites means that your validations were hardly +/- 3h (the standard for true satellite validation (Bailey & Werdell, 2006)…see general comments above)!  Also, which MODIS instrument did you use?  Both were aloft during these cruises.

P7 L207-239- It is not clear what exactly the PCA analysis was for (as related to the validation of the PIC volume method).  This should be stated in the lead sentence for that paragraph.

P8 L230-"Almost all" doesn't equal "solely"

P8 L233- What is the accuracy of the standard PIC algorithm?  How does this compare to the accuracy of the morphometric approach?  What is the absolute calibration?
standard for PIC?

P8 L234- Samples probably too small (volume-wise) to adequately sample forams and certainly not pteropods, other sources of Southern Ocean PIC .  See Trull et al. (Trull *et al.*, 2018) for data on this.  These can add significant PIC to the water.

P9 L261 I hope that in the Discussion, you compare this carbon per coccolith with other estimates (for which there are many).

P10 L301- And temporal scales, too!

P10 L307- Yes, but those are 1:1 clear-sky ship versus satellite comparisons using analytically-derived PIC concentrations.  Here, the authors are using a technique with unquantified errors (coccolith-morphometrics) which are not calibrated to analytically-derived PIC.  See general comments above.

P10 L310- Of course there are other particles!  You are only seeing the diatoms and coccolithophores and maybe armored dinos in the SEM since they would preserve well on the filter. But there are huge abundances of purely organic cels like *Phaeocystis*, picoplankton and nanoplankton (without any sort of mineral coverings to make them visible in the SEM).

P11 L335-336- The attribution of light intensity and iron limitation are meaningless here since no such data were taken in this study (to know if it is even remotely feasible).

P11 L351-356- You desperately need an error analysis.  You are presenting data from some 32 water samples (all surface, or some at depth?)  The level-3 satellite data (which originate over the top two optical depths in a weighted fashion) are averaging over large pixel sizes.  Did you pick just one pixel or the center pixel and surrounding 8 pixels?  Most of the matchups were done with monthly averages (which may indeed have only resulted from a single overpass given the cloudiness in this region.  A lot can happen to a population in one month! The potential errors on both sides are enormous.  And you have no standard (analytical PIC measurements) on which to judge any of these. See general comments.

P15 L454-455-  I'm sorry, but this is not a very definitive comparison, with no absolute standard analytical measurement.

P15 L458- Define "good".  Nowhere do you ever plot one versus the other, nor present statistics.

P15 L460- How do you know that the morphometric algorithm is also not biased, especially when there is no comparison to gold-standard analytical PIC measurements.

P15 L467- But no where did you present the actual abundance of diatoms.  What about the organic particles that you can't see with the SEM?

P15 L472- See Trull (Trull *et al.*, 2018) for elsewhere in the Southern Ocean.  They show varying influence of other larger calcifiers!

P15 L485- Regarding future research, you never defined the accuracy of the morphometric method, nor discussed the accuracy of the satellite method.

Fig. 1 legend- What do big dots and little dots mean.  Are the big dots the actual samples that you estimated coccolith morphometrics?

Fig. 3 legend- Are all these measurements from the surface?  3m depth?  I don't recall seeing this.  Please add error bars to the data when possible.  I'm not sure these colors will necessarily be visible to a color-blind person.  Please use correct color pallet.  I'm not sure "electric blue" is

a good color descriptor to a color-blind person.  What are the units of "tube width"? The triangle and diamond symbols are so small that one can't discern them.

Fig. 4 legend- Same comments as for Fig. 3

Fig. 5 -  I'm confused for panel (C) since it shows 38 samples just for panel C (New Zealand transect) and 16 samples for the Drake Passage transect.  These don't match your original numbers?

Fig. 6- panel b- confusion again over the numbers of samples in the New Zealand transect (here showing 13 samples) and Drake Passage transect (here showing 15 samples). Symbols in these figures are tiny!  Make them bigger.

Fig. 7- I do not understand how this PCA figure advances the goal of the paper.  Eliminate?

Fig. 8- Not sure a color-blind person will be able to interpret the colors in this.  Can you add error bars to this given a thorough error analysis?  What is "C-Calcita" referring to?  PIC?

Fig. 9-  I see diatoms in all four SEMs.  Did you quantify them in any way?   I see plated coccolithophores and coccoliths in panel c  (from south of the Polar Front).  What does this mean for the hypothesis?

Summary

In summary, there are so many unquantified errors in these measurements (for both coccolith morphometrics and satellite PIC) that this reviewer cannot recommend publication of this paper in its current form.   If it is to be published, it will require a major revision.

References

Bailey, S. W., & Werdell, P. J. (2006), A multi-sensor approach for the on-orbit validation of ocean color satellite data products, *Rem. Sens. Environ.*, *102*, 12-23.

Holligan, P. M., Charalampopoulou, A., & Hutson, R. (2010), Seasonal distributions of the coccolithophore, *Emiliania huxleyi*, and of particulate inorganic carbon in surface waters of the Scotia Sea, *Journal of Marine  Systems*, *82*(4), 195-205.

Holligan, P. M., et al. (1993), A biogeochemical study of the coccolithophore, *Emiliania huxleyi*, in the north Atlantic, *Global Biogeochem. Cycles*, *7*(4), 879-900.

Oliver, H., McGillicuddy, J., D. .J. , Krumhardt, K. M., Long, M. C., Bates, N. R., Bowler, B. C., Drapeau, D. T., & Balch, W. M. (2023), Environmental drivers of coccolithophore growth in the Pacific sector of the Southern Ocean *Global Biogeochem. Cycles*, *37*(11), https://doi.org/10.1029/2023GB007751.

Robertson, J. E., Robinson, C., Turner, D. R., Holligan, P., Watson, A. J., Boyd, P., Fernandez, E., & Finch, M. (1994), The impact of a coccolithophore bloom on oceanic carbon uptake in the northeast Atlantic during summer 1991, *Deep-Sea Research Part I*, *41*(2), 297-314, doi:10.1016/0967-0637(94)90005-1.

Trull, T. W., Passmore, A., Davies, D. M., Smit, T., Berry, K., & Tilbrook, B. (2018), Distribution of planktonic biogenic carbonate organisms in the Southern Ocean south of Australia: A baseline for ocean acidification impact assessment, *Biogeosciences*, *15*(1), 31-49, doi:10.5194/bg-15-31-2018.

Young, J. R., Poulton, A. J., & Tyrrell, T. (2014), Morphology of Emiliania huxleyi coccoliths on the northwestern European shelf - Is there an influence of carbonate chemistry?, *Biogeosciences*, *11*(17), 4771-4782, doi:10.5194/bg-11-4771-2014.

Young, J. R., & Ziveri, P. (2000), Calculation of coccolith volume and its use in calibration of carbonate flux estimates, *Deep-Sea Research Part II: Topical Studies in Oceanography*, *47*(9-11), 1679-1700, doi:10.1016/S0967-0645(00)00003-5.

---

## Author Comment (AC1)

**Manuscript: egusphere-2023-2801 - response to reviewers**

Dear Jamie,

Thank you for providing an opportunity to respond to the detailed comments by two reviewers. Here, we provide our response to reviewer 1 comments, including the action that will be taken in a revised manuscript (much of which we have already done in preparation, such as the addition of better error analysis). The original reviewer comments are provided in black and italics; our responses are in red.

Best wishes,

Mariem Saavedra-Pellitero and co-authors

**REVIEWER #1**

*Synopsis: This paper evaluates the coccolithophores in 13 water samples from a transect south of New Zealand to Antarctica (Dec. 2004 to Jan. 2005) plus19 more water samples from a transect at the western end of Drake Passage (Feb.-March 2016). They made SEM images of the water samples, then calculated the PIC concentration of each sample using the morphometric properties of the coccoliths. Then they used level-3 satellite data to estimate PIC concentrations. These satellite data were binned to 4km-resolution and a mix of temporal binning (daily, 8-day average and one-month time-averaged satellite images but mostly one month binning). They then concluded that Subantarctic waters and waters north of the Polar Front, the NASA PIC algorithm worked reasonably well. South of the Polar Front, however, there were few coccolithophores yet the satellite observations still suggested elevated PIC. The authors suspect that other reflective particles than PIC were causing the PIC algorithm to be in error.*

**General comments**

*This paper deals with an interesting topic, the coccolithophores of the Southern Ocean and what appear to be false, high-concentrations of coccolithophores as derived by satellite when the coccolithophore concentrations were observed to be low in their 32 water samples. Unfortunately, the paper is lacking in some fundamental key aspects and this reviewer cannot recommend publication it its current form.*
We thank Reviewer 1(R#1) for their suggestions, which will help to notably improve this manuscript. We provide our response to each comment below.

*The key shortfalls are: (1) the NASA PIC algorithm measures the concentration of suspended PIC. Nowhere in the paper did the authors ever actually measure PIC analytically in order to judge the errors in the satellite approach,*
Our study explores whether PIC estimates derived from coccolithophore data correlated with satellite-derived PIC trends in our area of interest. Due to the lack of in situ measurements of PIC, the aim of this study was to identify broad temporal and spatial trends rather than to provide a precise validation of satellite data against analytically measured PIC in the field. Level 3 satellite-derived PIC data was chosen for the simplicity of its access and processing, but thanks to R#1's suggestions we will now include Level 2 satellite-derived PIC data.

*(2) the authors "measured" PIC using an approach based on the morphology of coccoliths and the numbers of coccoliths observed on coccolithophore cells under the SEM, calculating PIC volume. Nowhere did they use analytically-derived PIC estimates to examine the error of this SEM-based approach,*

We do not attempt to analytically measure PIC in this study and do not claim to have done this. This aspect of data collection was not the focus of the scientific cruises, which mainly focussed on sediment recovery, not on plankton studies. Our aim in this study is to estimate (not measure) PIC using coccolithophore abundances and in particular biometric measurements on the dominant coccolithophore species *Emiliania huxleyi*, and to compare it with PIC obtained from satellite remote sensing, which was used as a proxy for coccolithophore carbonate surface production (e.g., Balch et al. 2005). In a revised version of the manuscript, we will include standard deviation and errors in the text and on figures for both, coccolith PIC estimates and satellite derived PIC values.

*(3) the remote sensing approach used here was inadequate for estimating PIC in these Southern Ocean waters,*
The purpose of our study was not to measure PIC concentrations in the Southern Ocean waters for the periods of interest but to explore whether PIC estimates derived from coccolithophore data correlated with satellite-derived PIC trends in our area of interest over the period of interest.

*(4) their conclusion that the PIC algorithm does not perform well south of the Polar Front leads them to hypothesize that something else is causing the error. They hypothesized in the abstract that the reflectance signal from these waters was "due to the prevalence of small opal particles or unknown highly reflective particles (such as Phaeocystis aggregations) or suspended sediment " and later in the paper, they mention other possible sources of errors such as microbubbles, floating loose ice or high concentrations of other particulate matter such as glacial flour", which could cause errors. Unfortunately, barely any evidence was provided for these possibilities. I will elaborate on each of these points below.*
We agree with R#1 that our conclusions in relation to this one part of the study are somewhat speculative. However, considering the micropalaeontological / nannofloral nature of this manuscript, we can only hypothesise what could happen south of the Polar Front by citing published scientific literature, because we do not have enough evidence for that.
In the new version of the manuscript, diatom data will be added to Figures 3 and 4 as an additional line of evidence to support our conclusions.

**Analytical PIC measurements**

*The gold-standard for measuring PIC in the ocean uses analytical measurement of either particulate Calcium (using any of several methods: particulate calcium analyses using flame photometry or optical emission spectrometry, or carbonate carbon measurements using coulometry). Any of these techniques would have been ideal to use to absolutely validate the PIC concentrations with quantifiable accuracy and precision.*
This point is of course absolutely correct. However, we do not aim to absolutely validate satellite-derived PIC concentrations in this study - we are comparing broad temporal and spatial trends based on PIC estimated from coccolithophores and satellite-derived PIC.

**PIC volume measurements using SEM**

*The technique that these authors used (Young & Ziveri, 2000), involves SEM imaging of filtered coccolithophores for the multiple morphotypes of E. huxleyi (the most abundant coccolithophore species in these waters). Using this laborious technique, they quantified the following morphotypes for coccolithophores: Type A, Type A over-calcified, Type B, Type B/C, Type C, Type O single layered and Type O double-layered) as well as a few other coccolithophore species (Young & Ziveri, 2000). These measurements involve (for hundreds of cells) quantifying the length of the major and minor axes of the coccoliths, number of t-elements, ray width, tube width, and a shape factor. Each of these measurements will have an error, which would compound to the overall error of the coccolith volume, hence the PIC*

*per cell. No such error bars were ever provided for the PIC measurements of coccolith morphometrics from their 32 samples. Moreover, it was critical to provide the accuracy and precision of the PIC morphometric measurements, measured against calibrated, analytical PIC measurements. As for the influence of diatoms in the Southern waters causing error in the satellite measurements of PIC, the only evidence was shown is four SEM images that contained diatoms.*

We will add standard deviations and errors (following Rigual-Hernandez et al., 2020) for coccolith measurements to provide an idea of the accuracy. We also consider different *E. huxleyi* morphotypes, which, as shown by Poulton et al. (2013), is important because it is assumed there is a ~50 % lower content of coccolith calcite for the B/C morphotype of *E. huxleyi* compared to the A morphotype (~0.015 or 0.033-0.035 pmol C coccolith [Poulton et al., 2010, 2011]). Including different morphotypes depicts a more accurate picture and it lowers the error in the estimations of coccolith calcite according to Young and Ziveri (2000). Regarding the diatom data, it has been previously published in Malinverno et al. (2016) and Cardenas et al. (2018). We will also include this data in the revised version of this manuscript.

*They never presented the abundance or biomass data of diatoms present in the samples, (nor an estimate of the biogenic silica associated with them, for that matter). It would have been nice to know some of the other features of these waters, too, such as chlorophyll concentrations, POC concentrations, TSS (total suspended sediments) to tease out what the mystery particles were causing the discrepancy.*

We will add diatom data and MODIS-derived chlorophyll concentration (mg m-3) to figures 3 and 4. The other data mentioned does not exist for these two transects unfortunately.

**Remote sensing approach**

*The authors used questionable remote sensing techniques to derive the PIC concentration. It is not clear exactly what they did here and more information is needed to assess this. They did say that their satellite PIC estimates were based on Level 3 data, highly binned satellite data in space and time. For example, in creating Level 3 data, daily, one kilometer resolution measurements would have been aggregated and averaged into 4km pixels.*

We thank R#1 for this comment. MODIS L3 PIC is created by NASA Ocean Biology Processing Group (OBPG). We downloaded daily, 8-daily and monthly MODIS L3 PIC for the area and period of interest, and extracted the value of the pixels surrounding the coordinates of the sampled location. This is described in lines 192-200 of the manuscript, however, we will edit those lines with the new data for clarity.

*In addition, using data at 4km resolution could likely have included multiple overpasses over a day, especially given the high latitude. Multiple overpasses mean the radiances are associated with multiple azimuthal and nadir-viewing geometries. Why is this a problem? For algorithm development, one should always use Level 2 data. These data are recovered from a single overpass, first calculated from unprocessed instrument data (Level 0), to Level 1A data (reconstructed, unprocessed instrument counts), converted to calibrated water-leaving radiance data (Level 1B) at a given solar zenith and nadir viewing angle and those radiances are finally processed to geophysical variables as Level 2 data (e.g., PIC concentrations using the PIC algorithms at hand).*

We thank the reviewer for this insightful comment and for their advice to use MODIS L2 PIC for this study. As suggested, we will now incorporate L2 satellite-derived PIC in figures 3 and 4 for our comparisons with coccolithophore-PIC estimates. We find that this has no major impact on our findings; in fact, the L2 PIC data match even better than when we were previously using L3 data.

*Using already-binned, level 3 data can lead to spurious results for relating radiances to PIC due to the already-mentioned bin-averaged data, likely collected from multiple overpasses, and potentially applying an arithmetic average when the variables are actually log normal in distribution (requiring a geometric average) or vice versa. Algorithm validation should always be done with Level 2 data as per Bailey and Werdell (2006). There's too much binning that goes on to get from L2 to L3 to do an accurate comparison. It looks like, due to abundant cloudy skies in the Southern Ocean, these authors had to use monthly composites for the most part, to get the satellite PIC data to match with their ship samples. Bailey and Werdell (2006) recommend overpasses within ±3 hours of field observations (not even ±24 hours), in order to properly validate any ocean colour-derived variable. Here, the monthly level-3 composites would have been good to a temporal resolution of ±15d for the most part, hardly a "coincident" matchup! This is a long temporal difference to validate against. One can go from a phytoplankton bloom to background phytoplankton concentrations in just a few days, especially when grazers and viruses are devouring them! An error propagation calculation is essential, for all the measurements described in this paper (satellite, PIC volume), in order to assess the true errors.*

As mentioned above, we will now use L2 satellite-derived PIC in the revised manuscript as suggested by R#1. We acknowledge the comments about the importance of temporal resolution and the ideal use of narrow time windows for temporal coincidence when comparing satellite-derived PIC and coccolithophore PIC estimates. However, as R#1 noted, we did not use daily satellite-derived PIC data because of the gaps in the record caused by frequent cloud cover in the Southern Ocean. This is why we used 8-day and monthly composites, as they provided more complete spatial and temporal coverage for our analysis. Even when using L2 data, there are still data gaps as shown in the following Heatmaps we have created (Figures R1 and R2). We will elaborate on this limitation in our discussion and assess how these data gaps might affect the observed trends and correlations.

[Figure]

Figure R1: Heatmap showing the L2 PIC (mol m-3) for the New Zealand transect. Dates are shown in the X axis and stations (from North -up- to South -down-) in the Y axis. Note that the values on the labels are not logarithmic, but the colour scale is.

[Figure]

Figure R2: Heatmap showing the L2 PIC (mol m-3)  for the Drake Passage transect. X axis shows Dates are shown in the X axis and  stations (from North -up- to South -down-) in the Y axis. Note that the values on the labels are not logarithmic, but the colour scale is.

***Causes for PIC error in the satellite approach south of the Polar Front***

*On both transects, the authors saw a drop-off in the PIC volume calculations south of the Polar Front. However, the possible reasons are numerous, such as dense diatom populations, dense Phaeocystis aggregations, floating ice crystals, bubbles or other suspended sediments, to name a few. Yes, they saw some diatoms in these waters south of the Polar Front (see their Fig. 9c and d) but upon close inspection, there were diatoms in the waters from north of the Polar Front, too (see their Fig. 9a and 9b). They also saw coccolithophores in their SEM from south of the Polar Front (see Fig. 9c). Nonetheless, they ultimately concluded that diatoms must be responsible for this error. Note, the SEM technique would never have visualized soft-bodied Phaeocystis in an identifiable form, ice crystals or bubbles, even if they were present.*

It is correct that we saw some diatoms south of the Polar Front, not only in the four images included in our original submission but also in previously published studies. The (quantified) diatom data is available in Maliverno et al. (2016) for plankton samples and Cardenas et al. (2018) for surface sediment samples. We will add these data in figures 3 and 4 (but more details can be found in the original sources). We are also aware that *Phaeocystis* aggregations, ice crystals or bubbles would not be visible in our samples. We raise these as a possible explanation based on previously published work (e.g., Balch et al., 2011; Balch, 2018; Balch and Mitchell, 2023), but this is not something we are able (or aiming) to address fully in this paper.

*Suspended lithogenic sediments would have been visualized under SEM, however. Finally, foramifera can be abundant PIC sources in these Southern Ocean waters, too, contributing*

*~50% of the PIC in waters at 55-60oS (Trull et al., 2018) but they aptly point out that due to the optical properties of large PIC particles (they don't scatter light much per unit PIC), the satellite might not see their reflectance. But this still could be a source of error in their PIC volume calculation (which included only coccolithophores, not foraminifera or fragments of foraminfera).*

We thank R#1 for bringing up this foraminifera point. Certainly, there is PIC produced by foraminifera. However, it is well established that coccolithophores (including detached coccoliths) are responsible for the majority of PIC backscatter (Balch et al., 1996; Balch and Mitchell, 2023). This is mentioned in the introduction.

Balch, W.M., Kilpatrick, K., Holligan, P.M., Harbour, D., Fernandez, E., 1996a. The 1991 coccolithophore bloom in the central north Atlantic. II. Relating optics to coccolith concentration. Limnol. Oceanogr. 41, 1684–1696.

Balch, W and Mitchell, C. 2023 Remote sensing algorithms for particulate inorganic carbon (PIC) and the global cycle of PIC
Earth-Science Reviews, 239 (2023), p. 104363

*This paper desperately needs an error analysis so that the authors can put error bars on the PIC estimates that appear in the figures and tables.*
We agree and will add error analysis to all figures and tables in the revised version.

***Specific comments:***
*P1 L23: Spelling "Coccolith"*
We have corrected the spelling mistake

*P1 L26: Which satellite product was used? What sensor?*
This information is stated in line 206 of the manuscript, in section 3.4. In the new version L2 data will be used. The information is included also in the Methods section.

*P2 L47: They should also cite the earlier references who were really the first ones to show this (Holligan et al., 1993; Robertson et al., 1994).*
Those references will be added.

*P3 L89: change word "concentrations" to "measurements". Next sentence, change "Most" to "many" since there are a significant number of subpolar studies that enumerated free coccoliths.*
Those two changes will be made

*P3 L94: See also (Young et al., 2014)*
This reference will be added

*P4 L99: The Oliver et al paper is now published (Oliver et al., 2023) and it included non-bloom areas, too, I believe.*
The citation will be updated.

*P4 L104: Does aim #1 include both plated coccolithophores as well as detached coccoliths?*
Coccolith morphometrics were always performed on coccospheres. The number of detached coccoliths/L (plus coccospheres/L) was only considered in the estimates for the New Zealand transect for the PIC estimates (see Malinverno et al., 2015). This will be mentioned in revised sections 3.2 and 5.1). Saavedra-Pellitero et al. (2019) only considered cells/L (detached coccoliths/L numbers for the Drake Passage were not available). That is why this aim is more general and we decided to keep it as it is.

*P4 L117 and occurrences henceforth in the manuscript: In the field of physical oceanography, salinity data should not be reported with units of ‰.*
The units (‰) will be deleted

*P5 L135: Cite also (Holligan et al., 2010)*
We believe that this refers to the original L132-133, so we will add the reference there).

*P5 L143: R/V Italica should be italicized.*
We will italicise *R/V Italica* and *Polarstern*, also in the figure captions.

*P5 L144: Is XX referring to the roman numeral or is this some sort of place-holder that needs to be filled in?*
XX refers to the roman numeral, so this will remain unchanged.

P5 L147-154: The authors are critical of how few stations are available in NASA's remote sensing validation of the PIC algorithm (n=42; noting that these are clear-sky matchups, using L2 satellite data within +/- 3 hours of an overpass (see general comments above). It should be noted, too, there is very strong under-sampling in this study, with a total of 32 water samples from which the paper is based, and these samples are based on Level 3, binned data which should not be used in satellite validation.
We have not claimed to have performed any satellite validation in this study, we just provided the validation statistics from NASA's Ocean Biology Processing Group (OBPG) PIC validation products. We will also now use L2 satellite data, as suggested previously.

*P6 L168: There are some details the authors might elaborate on. A SEM only views one side of a coccosphere. What about the coccoliths one can't see? Is this taken into consideration with the shape factors? It looks like they approximate the coccolith volume in equation 1 as a cube (L3), why not a sphere?*
To estimate the number of *E. huxleyi* coccoliths per coccosphere we counted the visible ones (half coccosphere) and multiplied by two (see Table S4). This information will be added to the revised manuscript. For other species, generic numbers were taken from Young and Ziveri (2000). All this information is available in Table 1.
Equation 1 was just taken from Young and Ziveri (2000) and it has been broadly used in the literature (even in papers published in Biogeosciences, such as Rigual Hernández et al., 2020). A full discussion of this is beyond the scope of this study.

*P6 L186: For those not familiar with the Young and Ziveri method, could you say what "tube" you are talking about?*
L186 refers to the *relative tube width'* (an index calculated using equation 2) not the *tube width* (the latter being the tube between central area and rim or "T-elements", see figure 2 or R3 for clarity). Considering that Young et al. (2014) have shown that the degree of calcification of a coccolith appear to broadly co-vary with the *relative tube width*, we will change the *relative tube width'* for the original index (*relative tube width*; Figure R3) - which makes more sense - in the new version. We therefore modified Figures 3, 4 and 6 accordingly. We will also refer to Figure 2 at the end of that sentence, which clearly shows the measurements involved in equation 2 in different SEM pictures.

[Figure]

$$\frac{\text{relative}}{\text{tube width}} = \frac{2 \times \text{tube width}}{\text{coccolith width}}$$

Figure R3. Morphometric parameters measured by Young et al (2014) and the equation used to calculate the *relative tube width.*

*P7 L192: What fraction of your samples used daily, 8-daily or monthly PIC values? Was there a difference on how well the validation agreed when you used the shorter time scale validations?*
As discussed in previous comments, we did not validate the satellite-derived PIC with in situ PIC concentration.

*You should state this. I would suspect that since you show monthly images in your Figure 1, then most of your comparisons were done with monthly images. Is this correct? Your comment at line 204-205 would suggest that the number of daily match-ups was extremely low. Your table 2 lists these out but for the actual validation, if you had a daily, an 8-d and a monthly measurement, which did you use? Please explicitly state this! Using 8-d composites or monthly composites means that your validations were hardly +/- 3h (the standard for true satellite validation (Bailey & Werdell, 2006)…see general comments above)! Also, which MODIS instrument did you use? Both were aloft during these cruises.*
We use monthly PIC rasters in Figure 1 maps as background. As stated in line 220, we used both weekly (8-day) and monthly satellite-derived PIC data in the study. We did not perform validation of the satellite-derived data, only compare the coccolith PIC estimates with the weekly and monthly values. The comparisons were graphed in figures 3 and 4, and will be updated with L2 data in the new version.

*P7 L207-239- It is not clear what exactly the PCA analysis was for (as related to the validation of the PIC volume method). This should be stated in the lead sentence for that paragraph.*
We will delete the PCA so this comment does not apply anymore.

*P8 L230-"Almost all" doesn't equal "solely"*
We will delete "solely" since it was misleading.

*P8 L233- What is the accuracy of the standard PIC algorithm? How does this compare to the accuracy of the morphometric approach? What is the absolute calibration? standard for PIC?*

NASA Ocean Biology Processing Group (OBPG) validates PIC retrievals performed with available matchups returns mean bias of 0.30577 and mean absolute error of 4.00304 (both values calculated after applying the log10 transformation to the PIC values) (https://oceancolor.gsfc.nasa.gov/data/reprocessing/r2022/aqua/). This information will be included.

*P8 L234- Samples probably too small (volume-wise) to adequately sample forams and certainly not pteropods, other sources of Southern Ocean PIC . See Trull et al. (Trull et al., 2018) for data on this. These can add significant PIC to the water.*
R#1 is right. Unfortunately our samples are too small to check other possible sources of PIC. We added this information in the discussion.

*P9 L261 I hope that in the Discussion, you compare this carbon per coccolith with other estimates (for which there are many).*
We do compare our data to other estimates in the discussion.

*P10 L301- And temporal scales, too!*
The temporal scale will be added.

*P10 L307- Yes, but those are 1:1 clear-sky ship versus satellite comparisons using analytically derived PIC concentrations. Here, the authors are using a technique with unquantified errors (coccolith-morphometrics) which are not calibrated to analytically-derived PIC. See general comments above.*
We will add errors and standard deviations to our datasets.

*P10 L310- Of course there are other particles! You are only seeing the diatoms and coccolithophores and maybe armored dinos in the SEM since they would preserve well on the filter. But there are huge abundances of purely organic cels like Phaeocystis, picoplankton and nanoplankton (without any sort of mineral coverings to make them visible in the SEM).*
We will delete "limited", make it clear that we are talking about our study area only, and add a reference here to emphasise that there are indeed other biogenic particles.

*P11 L335-336- The attribution of light intensity and iron limitation are meaningless here since no such data were taken in this study (to know if it is even remotely feasible).*
Since this is simply a case of speculating in the discussion, we will modify this statement.

*P11 L351-356- You desperately need an error analysis.*
We will add errors and standard deviations to our datasets

*You are presenting data from some 32 water samples (all surface, or some at depth?)*
This has been indicated in the methods and figure captions already, but we will add it here also.

*The level-3 satellite data (which originate over the top two optical depths in a weighted fashion) are averaging over large pixel sizes. Did you pick just one pixel or the center pixel and surrounding 8 pixels? Most of the matchups were done with monthly averages (which may indeed have only resulted from a single overpass given the cloudiness in this region. A lot can happen to a population in one month! The potential errors on both sides are enormous. And you have no standard (analytical PIC measurements) on which to judge any of these. See general comments.*
We use the value of the pixel that corresponds to the observation location. As already mentioned in previous comments, because this study does not aim to measure in situ PIC concentration values and because there are big gaps in the satellite-derived PIC record, we

did not perform a precise temporal matchup. Instead, our approach is to compare both sources of PIC to see if their spatial and temporal trends were similar.
However, will also now use L2 satellite data, as previously mentioned.

*P15 L454-455- I'm sorry, but this is not a very definitive comparison, with no absolute standard analytical measurement.*
We will tone this statement down.

*P15 L458- Define "good". Nowhere do you ever plot one versus the other, nor present statistics.*
A plot with the correlation coefficient will be added to the supplementary material.

*P15 L460- How do you know that the morphometric algorithm is also not biased, especially when there is no comparison to gold-standard analytical PIC measurements.*
This is correct, we are not able to judge if the algorithm is not biassed. We will make this more obvious in the conclusions.

*P15 L467- But no where did you present the actual abundance of diatoms. What about the organic particles that you can't see with the SEM?*
We will add diatom abundance to Figures 3 and 4 and mention the potential influence of other biogenic particles in the text.

*P15 L472- See Trull (Trull et al., 2018) for elsewhere in the Southern Ocean. They show varying influence of other larger calcifiers!*
We are grateful for the reference provided. We were just referring to the potential contribution of coccoliths to the PIC in the previous version of the manuscript, but it is true that larger calcifiers and fragments of larger carbonate-forming organisms could be important. This will be mentioned in the updated version.

*P15 L485- Regarding future research, you never defined the accuracy of the morphometric method, nor discussed the accuracy of the satellite method.*
The morphometric method of Young and Ziveri (2000) was assumed to be the most advantageous for our data set, since both the taxonomic analysis and the morphometric measurements of the coccoliths had been performed using a SEM. This is undoubtedly the most accurate approach for identifying coccoliths to a morphotype level, as well as for measuring the coccoliths length, allowing us to produce a consistent coccolith biogeochemical dataset without other errors linked to the preparation of new samples and to the coccolith-calcite calibration, both required for the optical/birefringence analysis, which would represent the alternative for the calculation of the coccolithophorid PIC.

NASA Ocean Biology Processing Group (OBPG) validates L2 PIC retrievals performed with available matchups returns mean bias of 0.30577 and mean absolute error of 4.00304 (both values calculated after applying the log10 transformation to the PIC values) (https://oceancolor.gsfc.nasa.gov/data/reprocessing/r2022/aqua/).

We will edit the text so that we are able to better define accuracy as stated above in the revised version.

*Fig. 1 legend- What do big dots and little dots mean. Are the big dots the actual samples that you estimated coccolith morphometrics?*
This information will be added to the caption: "Large dots indicate samples in which biometries on *Emiliania huxleyi* were performed, and small dots where coccolithophore census were available".

*Fig. 3 legend- Are all these measurements from the surface? 3m depth? I don't recall seeing*

*this.*

The information was provided in Sections 3.1.1 and 3.1.2 but will also be added to the figure 3 and 4 captions.

*Please add error bars to the data when possible.*
*I'm not sure these colors will necessarily be visible to a color-blind person. Please use correct color pallet. I'm not sure "electric blue" is a good color descriptor to a color-blind person. What are the units of "tube width"? The triangle and diamond symbols are so small that one can't discern them.*
We will add one standard deviation to our PIC estimates as in Rigual-Hernandez et al (2020).
Symbols will be enlarged, colours will be modified and figures checked here:
https://www.color-blindness.com/coblis-color-blindness-simulator/
The caption will be modified accordingly
Relative tube width' has no units. See Young et al. (2014) for further information.

*Fig. 4 legend- Same comments as for Fig. 3*
Symbols were enlarged, colours were modified and figures were checked here:
https://www.color-blindness.com/coblis-color-blindness-simulator/
The caption was modified accordingly

*Fig. 5 - I'm confused for panel (C) since it shows 38 samples just for panel C (New Zealand transect) and 16 samples for the Drake Passage transect. These don't match your original numbers?*
This was not very clear previously, but the initial counts for the New Zealand Transect were performed in light microscope (58 samples), from those, a subset of 38 was analysed in SEM to distinguish the different morphotypes, and from those we picked 13 samples for the biometrics. This will be explained in the updated figure caption.
In the case of the Drake Passage, 17 were included, but two of the sampling points are really close, so it looks like there are 16. In 2 of the samples there were no coccolithophores, that is why there are fewer samples. This will be mentioned now in the figure caption.
We will also reduce the width of the bars, so all the sample locations are more visible

In doing this we discovered that we made a mistake calculating coccolith PIC (due to *Calcidiscus leptoporus* average length being 5.7 insead of 5), so all the figures will be updated (3, 4 and 5), as well as Table 1.

*Fig. 6- panel b- confusion again over the numbers of samples in the New Zealand transect (here showing 13 samples) and Drake Passage transect (here showing 15 samples). Symbols in these figures are tiny! Make them bigger.*
Some of the samples offshore Chile are really close to each other, and that is why it looks like there are fewer samples.

*Fig. 7- I do not understand how this PCA figure advances the goal of the paper. Eliminate?*
We will delete this.

*Fig. 8- Not sure a color-blind person will be able to interpret the colors in this. Can you add error bars to this given a thorough error analysis? What is "C-Calcita" referring to? PIC?*
This figure has been modified in order to address these points raised.
The legend will better specify what is what in the new version.

*Fig. 9- I see diatoms in all four SEMs. Did you quantify them in any way? I see plated coccolithophores and coccoliths in panel c (from south of the Polar Front). What does this mean for the hypothesis?*
This information will be added to figure 3 in the revised version.

*Detached coccoliths/L and coccospheres/L were both considered when estimating PIC for that transect. We will make this clear in the revised manuscript.*

**Summary**

*In summary, there are so many unquantified errors in these measurements (for both coccolith morphometrics and satellite PIC) that this reviewer cannot recommend publication of this paper in its current form. If it is to be published, it will require a major revision.*
We will add error assessments to our datasets as suggested by R#1. We hope this addresses these overall concerns.

**References**

*Bailey, S. W., & Werdell, P. J. (2006), A multi-sensor approach for the on-orbit validation of ocean color satellite data products, Rem. Sens. Environ., 102, 12-23.*
This reference will be added.

*Holligan, P. M., Charalampopoulou, A., & Hutson, R. (2010), Seasonal distributions of the coccolithophore, Emiliania huxleyi, and of particulate inorganic carbon in surface waters of the Scotia Sea, Journal of Marine Systems, 82(4), 195-205.*
This reference was already included in the previous version

*Holligan, P. M., et al. (1993), A biogeochemical study of the coccolithophore, Emiliania huxleyi, in the north Atlantic, Global Biogeochem. Cycles, 7(4), 879-900.*
This reference will be added

*Oliver, H., McGillicuddy, J., D. .J. , Krumhardt, K. M., Long, M. C., Bates, N. R., Bowler, B. C., Drapeau, D. T., & Balch, W. M. (2023), Environmental drivers of coccolithophore growth in the Pacific sector of the Southern Ocean Global Biogeochem. Cycles, 37(11), https://doi.org/10.1029/2023GB007751.*
This reference was already included in the previous version, but the paper was not yet accepted at the time of submission. The correct reference and citations will be updated.

*Robertson, J. E., Robinson, C., Turner, D. R., Holligan, P., Watson, A. J., Boyd, P., Fernandez, E., & Finch, M. (1994), The impact of a coccolithophore bloom on oceanic carbon uptake in the northeast Atlantic during summer 1991, Deep-Sea Research Part I, 41(2), 297-314, doi:10.1016/0967-0637(94)90005-1.*
This reference will be added

*Trull, T. W., Passmore, A., Davies, D. M., Smit, T., Berry, K., & Tilbrook, B. (2018), Distribution of planktonic biogenic carbonate organisms in the Southern Ocean south of Australia: A baseline for ocean acidification impact assessment, Biogeosciences, 15(1), 31-49, doi:10.5194/bg-15-31-2018.*
This reference was already included in the previous version

*Young, J. R., Poulton, A. J., & Tyrrell, T. (2014), Morphology of Emiliania huxleyi coccoliths on the northwestern European shelf - Is there an influence of carbonate chemistry?, Biogeosciences, 11(17), 4771-4782, doi:10.5194/bg-11-4771-2014.*
This reference was already included in the previous version

*Young, J. R., & Ziveri, P. (2000), Calculation of coccolith volume and its use in calibration of carbonate flux estimates, Deep-Sea Research Part II: Topical Studies in Oceanography, 47(9-11), 1679-1700, doi:10.1016/S0967-0645(00)00003-5.*
This reference was already included in the previous version

---

## Author Comment (AC2)

**Manuscript: egusphere-2023-2801 - response to reviewers**

Dear Jamie,

Thank you for providing an opportunity to respond to the detailed comments by two reviewers. Here, we provide our response to reviewer 2 comments, including the action that will be taken in a revised manuscript (much of which we have already done in preparation, such as the addition of better error analysis). The original reviewer comments are provided in black and italics; our responses are in red.

Best wishes,

Mariem Saavedra-Pellitero and co-authors

**REVIEWER #2**

**Summary evaluation and general comments**

*I think the study is valuable and, after some major revision, should be published. One overall issue is with the way they focus the study in the Title, Abstract and Intro goes into the weakest result where they do not succeed in resolving things so well. They clearly consider the greatest importance to be the comparison of the in-situ and satellite data, but I think they end up raising more questions in this sense than they succeed in resolving. A major deficiency is the lack of direct discrete measures of PIC by chemical analysis with to which to compare microscopy-based and satellite estimates of PIC. Without such data, they really can't complete a proper comparison. I recognize that it is not always trivial to get or process such samples, and I see some other points of value.*

*The strongest findings are that*

*a) coccolithophores are not contributing significantly to the apparent high PIC estimated by satellite in the Antarctic Zone, but are important further north;*

*b) they strengthen what we know of the biogeographic patterns in the high latitude southern hemisphere, picking two very important but quite distinct part of the circumpolar circulation;*

*c) there are uncertainties and disagreements between methods for estimating coccolith-PIC using SEM and or polarized microscopy and they provide information that allows evaluating that disagreement.*

*The first of those three points, that the AZ satellite signal assigned to PIC is not from coccolithophores, is both one of the most important results, but it is also the only part where they do a strong comparison between satellite and microscope-based approaches. In the rest, they do not complete the comparative analysis with a statistical comparison of the different approaches.*

*Consequently, my recommendation is that the authors should be encouraged to submit a major revision, and I suggest to them to change their focus to the points where they generate robust results that are nevertheless valuable.*

We thank Reviewer 2 (R#2) for their helpful suggestions that will lead to a much improved manuscript. We provide our response to each point below.

***Major specific comments:***

*Lines 101-106 and later (generally). This study seems to combine previously published data with new analysis, specifically with new analysis that relies on samples which were analyzed more deeply. It would help to cite that or those previous studies from these transects in this paragraph. I think early in the Methods there should be a first table defining where the samples come from, including noting in which previous studies they have been used in.*

A new table will be added that includes this information. Also, we will highlight the already published datasets in figures 3 and 4.

*Lines 191-206: 3.3 Satellite-derived PIC data*

*I think it is interesting to compare in situ data to the output distributed from the NASA group, but then I am not sure what exactly they did. Did they compare a single pixel to each discrete sampling?*

Bailey and Wedell (2006) outline the method used by NASA Ocean Biology Processing Group (OBPG) for validating operational ocean colour missions. According to the authors, they would have compared the mean of a 5x5 window, centred on the in situ location. The in situ data utilised was extracted from SeaBASS and the Aerosol Robotic Network - Ocean Color (AERONET-OC). Validation results for MODIS-Aqua PIC are available on NASA's Ocean Color Portal (https://seabass.gsfc.nasa.gov/search/?search_type=Perform%20Validation%20Search&val _sata=1&val_products=9).

*Beyond showing later that there is a region of very high satellite signal with low or absent coccolithophores, I suspect this sampling is not really sufficient to do a true comparison of the satellite vs discrete measures.*

R#2 is right to point out that the sampling is not sufficient to do a true comparison, particularly because the discrete measurements do not provide analytically-measured concentrations of PIC. Additionally, the satellite-derived PIC data is scarce due to cloud cover. See also reply to Reviewer 1 (R#1) as well as figures R1 and R2 in that response.

***Section 4.1.***

*Lines 233-237: "Weekly and monthly MODIS-derived PIC at the sampling locations consistently overestimate PIC values… with respect to in-situ values calculated from coccolith mass" and "there is a relatively good agreement in the latitudinal satellite and coccolith-PIC trends in the SAZ and PFZ" and again at lines 292-293: "In the studied transects, the calculated coccolith-PIC and the satellite-derived PIC trends show quite good agreement in the SAZ and PFZ,"*

*This is a possibly highly subjective, evaluation of correspondence has been made. What is "good agreement"? Sometimes when coccolith-estimated PIC is low, satellite-estimated PIC is also low, and when coccolith-estimated PIC is higher, satellite-estimated PIC is also higher, but, even in the SAZ and PFZ they do not track precisely. I do not think they could track precisely even if all PIC was from coccoliths and the estimations were completely faithful, because one is a very discreet estimate at a single sample in a single Niskin bottle, @and the satellite value is an average over at least many days. So they can't correlate perfectly. Nevertheless, to make the statement it would be nice to see a direct analysis of the correlation between coccolith-PIC (calculated) and satellite-estimated PIC. That is, I'd like to see figure, combining both datasets, plotting one against the other.*

This point was also raised by R#1. This analysis will now be included in the supplementary material.

*Also, the phrasing could be more careful in lines 233-244, because "consistently overestimate PIC values" is only correct if one knows that most PIC comes from coccolithophores instead of other sources, which might not be true. Also, it is not clear if there is a way of knowing the error on the coccolithophore PIC estimation.*

We will reword these sentences and add information about potential errors relating to other sources of PIC as well as standard deviation in the text, figures and tables. This was also raised by R#1.

*There are several reasons why the estimates could not agree. Here are just some principal possibilities:*

*Perhaps the satellite estimates in fact are right, but a lot of coccolith PIC is in the form of coccolith fragments that are too fragmented to be recognized and counted, so microscopy estimates would be lower than the "true" value*

*Perhaps there is another PIC producer (although that possibility was implicitly considered in the Intro)*

*Perhaps the assumptions in the coccolith-PIC calculations are actually wrong (e.g., the shape factor is different than estimated, or the number of coccoliths per cell assumed is wearing)*

*In the Discussion, the authors do touch on some of these and other possibilities, but I would like to suggest they might try a modest reorganization so as to make it clear to go one by one through the possibilities to consider.*

We are very grateful for those suggestions. We will reorganise the discussion to incorporate these points.

*Line 270: "The relative tube width' (an index for calcification; Young et al., 2014), calculated using equation 2, varies from…"*

We will amend (now *relative tube width*).

*How do you deal with the issue that the "overcalcification" is variable? The tube width in the "overcalcified" forms tends to be very irregular, and sometimes the central area is covered nearly fully. This should be made explicit, perhaps with a new panel in Fig. 2 or a supplementary figure associated with the Methods section to show how this complicated situation (like the case in Fig. 6 bottom left) is dealt with.*

We agree that the calcification of the central area is variable, just as all measured parameters of the placoliths are highly variable. "Overcalcified" specimens were in general rather rare, but as we have assumed an average value for all other parameters, we have calculated a mean ks value that is higher for this morphotype than for the other morphotypes (see Table 1).

This data is available in the supplementary material submitted to Biogeosciences (Tables S1 and S2). It is also stored in Pangaea: https://doi.pangaea.de/10.1594/PANGAEA.964672 and https://doi.pangaea.de/10.1594/PANGAEA.964674 -but still under moratorium-.

We did not add any extra panel or supplementary figure, because the other morphotypes are more dominant (see cells/L in figures 3 and 4).

*I have some concerns about the PCA and how it is discussed.*

*First, the PCA does not seem to be very successful at separating the coccoliths based on morphometric parameters. Except for the overcalcified forms, which separate only based on tube width (needs to be clearer how that is really measured, as discussed above), the other forms overlap a lot. In fact, component 2 in the PCA, which correlates mostly with other morphometric characters quantified, does not appear to separate the morphotypes at all.*

*Some of the characters that have been used for separating types A and B are not analyzed, and might be quite difficult to analyze automatically (such as central area type, or whether the distal shield elements or straight or not). I think it is definitely valid to do and report the PCA, but it is important to discuss the fact that this analysis, using the current state-of-the-art quantification, seems to fail at what are visually quite striking differences is very notable. This suggests that perhaps it is time to try new approaches to distinguishing these types.*

Also following a comment from R#1, we will remove the PCA section from the revised version. We will incorporate the other very interesting points raised here in the discussion of the revised manuscript.

*Second, the issue also makes me wonder if we have independent estimates of how well the approaches of trying to estimate coccolith PIC by morphometry or by polarized microscopy work when comparing different morphotypes. It seems that, while there are many studies that directly quantity PIC in culture (either as acid-labile particulate carbon or as particulate calcium), it is hard to find any that also count the number of coccoliths to measure PIC/coccolith. It might be worth highlighting this need. I should mention, and not parenthetically, that I appreciated a lot the analysis they showed in Fig. 8, which goes directly to this point. I think their data suggests we are still lacking precision in the way we measure coccolith PIC, and I would hope that, while their study might not resolve it, at least it helps identify the problem.*

Good points, and we will elaborate further on this topic in the new version of the manuscript.

*Third, in lines 346-347 they say: "The PCA performed on the E. huxleyi morphometric dataset shows that those heavily calcified type A coccospheres occupy a relatively restricted ecological niche offshore of Chile" The PCA is only based on morphometric characters, regardless of the location or oceanographic conditions in which they are found. How can such a PCA, which does not include any environmental information, indicate anything about whether the ecological niche of one form is restricted or not?*

Good point. We know that just because previous published papers dealt with the different morphotypes and the environmental preferences (add citations here), but they were not included here. That is why we decided to (initially) focus on the PIC and leave the PCA out of the main story. Ultimately, and in response to a R#1 comment too, we have decided to remove the PCA aspect.

*For Figure 1, and the corresponding description in the Methods (lines 108-122), I could not tell if the white lines correspond to average positions of the fronts or to the positions the fronts occupied at the time of the study.*

We will include a modified figure that uses a colour blind friendly palette and make clear that they are average positions (based on Orsi and Harris, 2019).

**Minor technical and comments and corrections**

*Abstract: Eliminate page breaks within abstract*

It will be done

*Line 84: Perhaps reserve "concentration" for chemicals (such as PIC) and "abundance" for cell or coccolith numbers/volume. It's not strictly necessary, but it might help to be clearer*

It will be done

*Line 88-89 : " available coccolithophore concentrations" maybe clearer "available measurements of coccolithophore abundances"*

It will be done

*Line 98: "have targeted areas of coccolithophore bloom". Should be "blooms" (plural)*

It will be done

*172-173: "and modified for E. huxleyi according to the degree of calcification obtained for each morphotype (see Table 1)" How is this modification of KS according to calcification performed? Is there a reference?*

The different shape factors used were based on the identified morphotype following Young and Ziveri (2000): ks = 0.02 for morphotypes A and B/C and ks = 0.04 for morphotype A overcalcified. The shape factor for morphotype O (ks = 0.015) was introduced by Poulton et al. (2011) in a plankton study along the Patagonian Shelf for a morphotype with a central area described as an "open or thin plate" which the authors called type B/C but that we identified as morphotype O based on the published images and description of Hagino et al. (2011).

The dataset linked to this answer is available in the supplementary material submitted to Biogeosciences (Tables S1 and S2). It is also stored in Pangaea: https://doi.pangaea.de/10.1594/PANGAEA.964672 and https://doi.pangaea.de/10.1594/PANGAEA.964674 -but still under moratorium-.

The reference that compiles that information is Vollmar et al. (2022) and it has been included in the new version.

Vollmar, N. M., Baumann, K.-H., Saavedra-Pellitero, M., and Hernández-Almeida, I.: Distribution of coccoliths in surface sediments across the Drake Passage and calcification of Emiliania huxleyi morphotypes, Biogeosciences, 19, 585–612, https://doi.org/10.5194/bg-19-585-2022, 2022.

See also our response to R#1 for further details.

*Line 178: "Measurements of the distal shield diameters of Calcidiscus leptoporus the second most abundant species" Need a comma after "leptoporus"*

It will be done

*Line 265-266: "Note that this data is shown in Figures 3 and 4, but the coccolith-PIC was calculated in this work using equation 1 and the average lengths mentioned in Table 1 " Not clear.*

All the data is shown in Figure 3c and 4 c, but to estimate the coccolith PIC, the average length was considered. The sentence will be modified to make this clearer.

*Line 270: We observed that some coccoliths are clearly overcalcified (see Figure 5 for an example),…" Do you mean Fig. 6? Fig. 5 doesn't have any photos so can't see that observation.*

Yes, that is a mistake. We meant Figure 6. Thanks for spotting it.

---

## Referee Report (RR1)

Re-Review of Saavedra-Pellitero et al.; ms re-submitted to EGUSphere

**General comments:**

This reviewer acknowledges that the authors have made an effort to tighten-up the manuscript from the first version and scaled back their conclusions accordingly. I note that due to the tightening of this manuscript, the stated impact of this paper has also beenscaled back but, at least, it is more realistic now. After re-reading the paper, there are still a few issues to attend to that I will discuss below.

The title could be honed even more to state exactly what you are showing in this paper. I would suggest something like "Mismatch between coccolithophore-based estimates of particulate inorganic carbon (PIC) concentration and satellite-derived PIC concentration in the Pacific Southern Ocean." I think this is more realistic about what this paper is actually about.

The revised paper explains that they used different types of satellite data to estimate PIC remotely, (level 2 and level 3). Each type of satellite data has a different space and time averaging. They should state emphatically that they compared these different time and space averages of satellite data to the same ship data, (always collected at the same space/time scales). Note, these different space and time averages of the two data sets will affect any mismatch between the ship and satellite results, as well as the potential errors. Obviously, a lot can change in a coccolithophore population in +/- 8d or even +/- 1 month (see line 246) from a ship sample! It is still not really clear why they didn't simply use the level 2 data only (which would have less time and space errors but admittedly lower numbers of possible matchups due to the extreme cloudiness of the region). Did the authors ever use level 2 and level 3 satellite data in the same comparison with ship data? If so, those analyses cannot be used.

The estimates of PIC per coccolith that they cite regarding their model are huge. Beginning with lines 338-356 they discuss published estimates of calcite content per coccolith for *E. huxleyi*. They mention 0.015pmol per coccolith (Charalampopoulou et al., 2016), then they cite the author's own estimate of 4.64 pmol per coccolith as well as the calcite per coccolith used in this paper (of 1.66pg/coccolith [or at least I think those are their units!). This is an overall range in PIC per coccolith for *E. huxleyi* of over 300X! Using different PIC per coccolith values in their model could obviously affect the PIC concentration mis-match, too! The authors need to state why the calcite per coccolith values are so variable and how much of the magnitude of the mismatch is due to what value of the PIC per coccolith is used. (The 300X range in PIC per coccolith makes this reviewer suspect that there must be some mistake in the units of the calcite per coccolith being discussed in this paragraph).

They present discussion of the different morphotypes of *E. huxleyi* in this paper (lines 450-510). This reviewer is confused whether they are trying to connect the magnitude of the mismatch with the different morphotypes? The discussion seems more about simply presenting information about the different morphotypes that they encountered in the Southern Ocean. While interesting ecologically, does this have anything to do with the mis-match between SEM-based measurements of PIC and satellite measurements of PIC, the entire purpose of the paper? They could help this section by stating explicitly why they are presenting it at all. Only until I arrived

at the very end of the paper (line 536) did I see conclusion #4 stating "neither the slightly different carbonate masses nor the southward changes in morphotype composition had a decisive influence on the coccolith-estimated PIC, which is only determined by the abundance of *E. huxleyi* in this area".   This should be stated earlier in the discussion on morphotypes to tell the reader why you are including this morphometric data in the paper.

**Specific comments:**

L 1-2  See general discussion above regarding making the title more descriptive of the paper.

L 18-20  This sentence ("here we combine…") and the sentence in lines 21-23 ("We compare PIC estimates derived from…") are somewhat redundant.  I'd eliminate the first.

L 29  Change the words "coccolithophore data" to "coccolithophore-morphometric-based data"

L34 The word "zooplankton" comes out of nowhere.  Either eliminate altogether or state more carefully that you are referring to "calcifying microzooplankton."

L 41 Change "This process" to "Calcification"

L 54 Change "of the light backscattered…" to "of the total light backscattered.

L. 73  Understand the biological response to what?

L 94  Change "different calcification levels" to "degrees of calcification of coccolithophore cells".

L 102 "Here, we focus on the…"

L105  Change "outer bloom conditions" to "moderate bloom conditions"

L111-114 sentence beginning "The ACC…" This sentence is long-winded.  Please divide it in two.

L149  The model of SEM is a Tescan Vega, I believe.

L202  exchange word "reproducibility" for "precision".  Can you say anything about accuracy?

L223  Refer to coccolith-estimated PIC concentration not just PIC

L233 A zero SD could also mean there is only one measurement, right?

L239-242 Don't refer to general "data scarcity" but refer to the more specific "lack of cloud-free satellite images".  Also, don't say "to increase data availability" but "to increase the possibility of a ship-satellite match-up"

L 239-249 See general comments on how they mixed level-2 daily satellite data vs level 3, 8-day and monthly Level 3 satellite data.  Were Level 2 and Level 3 data used separately in different analyses or were they mixed in the same analysis?

L 339 Change "calcite weight" to "calcite mass per coccolith"

L341 State what the PIC per coccolith numbers were used by Poulton, as well as Rigual Hernandez.

L348 Change "we calculated PIC" to "we calculated total PIC concentration"

L379- As the handling editor mentioned, stay away from the word "validation" in this paper because you are not doing this!

L383  Too many uses of the word "estimated" in this sentence

L394  There is a need for both improved precision and accuracy!

L397- Get rid of sentence on validation of remote sensing data since this paper is not about that!

L435-  You refer to "subfossil diatoms".  Are these different from fossil diatoms.   Why not say "fossil diatoms in surface sediments"

Line 547- Instead of "zooplankton", do you mean calcifying micro-zooplankton like forams? Hard to imagine that your results will tell us anything about *Calanus*!

---

## Author Response (AR2)

Reply to the reviewers (2nd review) Saavedra-Pellitero et al.; ms re-submitted to EGUSphere

*Dear Authors,*
*Thank you for submitting your revised paper and for responding to the reviewer comments. Both reviewers have now read your revised manuscript and both have noted the improvements that you have made to the work. However, both of them have also identified some small improvements that can be made, where you have not quite addressed some of their original points.*

*So I would encourage you to submit a further revision and responses to their remaining points (the details of these are available in the second review reports that both have submitted). After you have done this, and assuming you address their remaining points, I envisage accepting the paper for publication in the journal.*

*I look forward to receiving you final revised paper and the accompanying responses to the second set of reviewer comments.*

*best wishes*
*Jamie Shutler*

**General comments:**

This reviewer acknowledges that the authors have made an effort to tighten-up the manuscript from the first version and scaled back their conclusions accordingly. I note that due to the tightening of this manuscript, the stated impact of this paper has also been scaled back but, at least, it is more realistic now. After re-reading the paper, there are still a few issues to attend to that I will discuss below.
Dear reviewer, thank you very much for carefully checking the new version of our manuscript and for your suggestions. Your time and effort are greatly appreciated.
Your comments definitely helped us to provide an improved version of the initial manuscript.

The title could be honed even more to state exactly what you are showing in this paper. I would suggest something like "Mismatch between coccolithophore-based estimates of particulate inorganic carbon (PIC) concentration and satellite-derived PIC concentration in the Pacific Southern Ocean." I think this is more realistic about what this paper is actually about.
We modified the title accordingly. The suggested title is more realistic about the contents of this piece of research.

The revised paper explains that they used different types of satellite data to estimate PIC remotely, (level 2 and level 3). Each type of satellite data has a different space and time averaging. They should state emphatically that they compared these different time and space averages of satellite data to the same ship data, (always collected at the same space/time scales). Note, these different space and time averages of the two data sets will affect any mismatch between the ship and satellite results, as well as the potential errors.

Obviously, a lot can change in a coccolithophore population in +/- 8d or even +/- 1 month (see line 246) from a ship sample! It is still not really clear why they didn't simply use the level 2 data only (which would have less time and space errors but admittedly lower numbers of possible matchups due to the extreme cloudiness of the region). Did the authors ever use level 2 and level 3 satellite data in the same comparison with ship data? If so, those analyses cannot be used.

Due to frequent cloud cover in the Southern Ocean, the availability of daily MODIS L2 data was limited, making it impossible to obtain matches for each field sampling location within the typical ±3-hour window (Bailey and Werdell, 2006) or on the same day. To address this lack of suitable data , we opted to aggregate all available Level 2 data for each location over the entire duration of the field campaign ± 7 days. We believed this allowed for a more robust comparison of coccolith-estimated PIC with satellite data, given the limited number of available daily observations.

Regarding the use of L3, we included these data to supplement the MODIS daily values due to their poor availability. MODIS L3 product's broader spatial and temporal resolution helped address L2 data gaps and ensured that the comparison of lab-measured PIC and satellite-derived PIC was meaningful despite the limitations of the satellite-derived data. The fact that L3 products are derived directly from L2 data ensures consistency in the underlying scientific methodology, despite the differences introduced by the spatial and temporal averaging in L3.

As previously explained, we did not conduct a formal validation in this study, but instead focused on comparing trends across coccolith-based and satellite-derived PIC. However, it is important to note that L3 products are derived directly from L2 data, remaining consistent with the underlying scientific methodology and retrieval processes (Scott and Werdell 2019). Therefore, while the aggregation in L3 can lead to smoothing and the loss of finer-scale variability, L2 validation results are generally applicable to L3 data (Scott and Werdell 2019).

In our study, we extracted values from both L2 (daily) and L3 (8-day and monthly) data for the same field locations. For L2, we followed the method outlined by Bailey and Werdell (2006), while for L3, we used the value of the enclosing grid cell. We then compared these point values, i.e. L2 extracted point values and L3 extracted point values, to the actual field measurements (figures 3 and 4). We were fully aware that this approach did not allow for a 'proper' match-up, due to the unavailability of L2 data. Therefore, we focused on comparing trends within what we considered a reasonable temporal window, understanding that while the data available did not allow for a strict validation, it still provided valuable insights into the general patterns of variability across both data types. This approach allowed us to assess broader trends while being mindful of the limitations and uncertainties introduced by the aggregation of L3 data.

Therefore, the use of both L2 and L3 data in our study was driven by practical considerations, such as the scarcity of available L2 data due to cloud cover and the need for more robust data for comparison with field measurements. The decision to include L3 data allowed for a more comprehensive comparison of satellite-derived PIC. While this introduces additional uncertainties, these variations are accounted for in our interpretation and do not diminish the overall consistency between L2 and L3 data.

We recognize the potential limitations in comparing in situ point measurements with satellite-derived 5x5 pixel averages, as this assumes a certain degree of spatial homogeneity that is not always present. However, this approach is standard in remote sensing studies (Bailey and Werdell, 2006), and despite inherent variability in the ocean, it remains a practical solution for comparing field-based and satellite data. In our study, this approach was further refined by aggregating L2 data over the full field campaign period ± 7 days to account for temporal variations, and we carefully considered the potential discrepancies when interpreting our results.

Given the unique challenges of working in the Southern Ocean, where cloud cover is frequent and can limit satellite data availability, we proceeded with the study using the available data. While the availability limitation of daily satellite observations posed limitations, we believe that aggregating the available L2 data and supplementing the comparison with weekly and monthly satellite observations (i.e. L3 data) was a reasonable approach to attempt a more comprehensive view of PIC variability across measuring techniques. These efforts were essential in overcoming the practical challenges of finding suitable remote sensing data in this region.

We have not combined L2 and L3 data in any single analysis. Each dataset (L2 and L3) has been analyzed independently, and the results for L2 and L3 data were not directly compared to each other for any given sample location.

Reference: Joel P. Scott and P. Jeremy Werdell. Comparing level-2 and level-3 satellite ocean color retrieval validation methodologies. Opt. Express 27, 30140-30157 (2019).

The estimates of PIC per coccolith that they cite regarding their model are huge. Beginning with lines 338-356 they discuss published estimates of calcite content per coccolith for E. huxleyi. They mention 0.015pmol per coccolith (Charalampopoulou et al., 2016), then they cite the author's own estimate of 4.64 pmol per coccolith as well as the calcite per coccolith used in this paper (of 1.66pg/coccolith [or at least I think those are their units!]). This is an overall range in PIC per coccolith for E. huxleyi of over 300X! Using different PIC per coccolith values in their model could obviously affect the PIC concentration mis-match, too! The authors need to state why the calcite per coccolith values are so variable and how much of the magnitude of the mismatch is due to what value of the PIC per coccolith is used. (The 300X range in PIC per coccolith makes this reviewer suspect that there must be some mistake in the units of the calcite per coccolith being discussed in this paragraph).
Thanks to the reviewer for spotting this mistake that we overlooked. It was indeed a typo in line 349: "4.64 pmol per coccolith" should be "4.64 pg per coccolith". We changed that and revised all the coccolith masses again to make sure the right units were used. We found a few typos that have now been fixed.

We also provided more details in section 3.2 (equation 1) regarding units.

In Figure 13 from Saavedra-Pellitero et al. (2019) we show this methodological difference in mass estimates, which is not 300X, but less than 1 degree of magnitude, but above all shows the same trend (this is also available in Figure 9 from this manuscript).

Due to the fact that some authors use pg and some pmol, we converted some of the published mass estimates to pmol (using the molecular weight of the calcite, i.e. 100 g/mol -section 3.2-) in the new Table 5 (which includes also new studies) and tried to stick to pmol in the discussion part to make it easier for the reader. Still mass in pg is sometimes mentioned if we refer to published papers, or in some of the figures (such as Fig. 9, which includes data from Saavedra-Pellitero et al., 2019).

[Figure]

*Figure 13 from SaavedraPellitero et al. (2019). Drake Passage latitudinal transects from east to west, showing box plot coccolith mass estimates (in pg): (a) this study, (b) transect at around 68° W from Charalampopoulou et al. (2016), and (c) transect at around 55–58° W from Charalampopoulou et al. (2016). Note that (b, c) have been calculated from Charalampopoulou et al. (2016). Outliers have been indicated with "+", and numbers on the x axes refer to the original station numbers. The approximate location of the Subantarctic Front (SAF) and Polar Front (PF) are shown as well as the year of the sampling.*

They present discussion of the different morphotypes of E. huxleyi in this paper (lines 450-510). This reviewer is confused whether they are trying to connect the magnitude of the mismatch with the different morphotypes? The discussion seems more about simply presenting information about the different morphotypes that they encountered in the Southern Ocean. While interesting ecologically, does this have anything to do with the mis-match between SEM-based measurements of PIC and satellite measurements of PIC, the entire purpose of the paper? They could help this section by stating explicitly why they are presenting it at all. Only until I arrived at the very end of the paper (line 536) did I see conclusion #4 stating "neither the slightly different carbonate masses nor the southward changes in morphotype composition had a decisive influence on the coccolith-estimated PIC, which is only determined by the abundance of E. huxleyi in this area". This should be stated earlier in the discussion on morphotypes to tell the reader why you are including this morphometric data in the paper.

We acknowledge this point raised by the reviewer. We tried to convey it by adding some information already in the Introduction (mentioning the morphological diversity of *E. huxleyi*), in the methods (see new table 2) and later on in the discussion, with the effect of *E.huxleyi* morphotypes on PIC (see new Section 5.1).

Note that we deleted most of the previous section "5.3 *Emiliania huxleyi* morphotypes" in order to focus more on the PIC and less on the ecology of the different morphotypes.

**Specific comments:**

L 1-2 See general discussion above regarding making the title more descriptive of the paper.
We modified the title.

L 18-20 This sentence ("here we combine...") and the sentence in lines 21-23 ("We compare PIC estimates derived from...") are somewhat redundant. I'd eliminate the first.
Done; we deleted that sentence.

L 29 Change the words "coccolithophore data" to "coccolithophore-morphometric-based data"
We changed it to "coccolith-based PIC"

L34 The word "zooplankton" comes out of nowhere. Either eliminate altogether or state more carefully that you are referring to "calcifying microzooplankton."
We deleted the word zooplankton

L 41 Change "This process" to "Calcification"
We changed it.

L 54 Change "of the light backscattered..." to "of the total light backscattered.
We changed it.

L. 73 Understand the biological response to what?
We reworded this sentence as: "Recent concerns about climate change have motivated the scientific community to focus on *E. huxleyi* as a target cosmopolitan species and in particular to differentiate it into different morphotypes"

L 94 Change "different calcification levels" to "degrees of calcification of coccolithophore cells".
We changed it.

L 102 "Here, we focus on the..."
We corrected that typo

L105 Change "outer bloom conditions" to "moderate bloom conditions"
We changed it.

L111-114 sentence beginning "The ACC..." This sentence is long-winded. Please divide it in two.

We split this sentence into two shorter ones.

L149 The model of SEM is a Tescan Vega, I believe.
We changed it to "SEM Tescan Vega".

L202 exchange word "reproducibility" for "precision". Can you say anything about accuracy?
We changed it.
We understand that accuracy measures how close results are to the known or published value. We compare the available measurements in the discussion part.

L223 Refer to coccolith-estimated PIC concentration not just PIC
We changed it (and used satellite-derived concentration in that sentence).

L233 A zero SD could also mean there is only one measurement, right?
Yes, that is correct, but we feel there is no real need to mention this in the manuscript.

L239-242 Don't refer to general "data scarcity" but refer to the more specific "lack of cloud-free satellite images". Also, don't say "to increase data availability" but "to increase the possibility of a ship-satellite match-up"
We changed those sentences.

L 239-249 See general comments on how they mixed level-2 daily satellite data vs level 3, 8-day and monthly Level 3 satellite data. Were Level 2 and Level 3 data used separately in different analyses or were they mixed in the same analysis?
See previous reply. We did not mix data from different levels. We used them independently. We reworded some sentences of this section to avoid confusion.

L 339 Change "calcite weight" to "calcite mass per coccolith"
We changed it.

L341 State what the PIC per coccolith numbers were used by Poulton, as well as Rigual Hernandez.
We added that information and rewrote part of that paragraph aiming to be more precise.

L348 Change "we calculated PIC" to "we calculated total PIC concentration"
We changed it.

L379- As the handling editor mentioned, stay away from the word "validation" in this paper because you are not doing this!
We agree with the Editor's suggestion, but this refers to the validation by the NASA Ocean Biology Processing Group (2023) and not by us, so we kept it.

L383 Too many uses of the word "estimated" in this sentence
We swapped estimating for calculating.

L394 There is a need for both improved precision and accuracy!
We reworded this sentence

L397- Get rid of sentence on validation of remote sensing data since this paper is not about that!

We deleted that sentence

L435- You refer to "subfossil diatoms". Are these different from fossil diatoms. Why not say "fossil diatoms in surface sediments"

Good suggestion, we changed it.

Line 547- Instead of "zooplankton", do you mean calcifying micro-zooplankton like forams? Hard to imagine that your results will tell us anything about Calanus!

We changed it and elsewhere in the manuscript

**Second revision of: Pacific Southern Ocean coccolithophore-estimated particulate inorganic carbon (PIC) versus satellite-derived PIC measurements**

The authors have made some important improvements to the manuscript. However, I think it still needs some revision. I think they have addressed the technical issues brought up in the first revision. Reviewer 1 in the first round had very tough but also very valuable criticisms based on the emphasis on comparing satellite PIC to coccolithophore PIC without the needed chemical PIC measurements to do a true validation. I think they have gone far in addressing these concerns. However, they still focus on comparing satellite PIC vs microscopic estimates of coccolith PIC, when I don't think that's a strong comparison (as they lack chemical measurements of total PIC as pointed out by both reviewers).

We appreciate the reviewer's comments and would like to clarify a couple of points. We are aware that unfortunately we do not have in-situ PIC concentration measurements, and that the comparison is made using carbonate in coccoliths, which has its own inherent limitations. Additionally, the satellite data was often compromised by cloud cover, which further impacted the precision of our comparisons.

With that in mind, we consider that the value of our work lies in its ability to highlight trends and patterns despite these limitations. Our goal has never been to produce a 'proper' match-up with PIC measurements, but rather to provide a qualitative, comparative analysis that can offer insights into broader trends. We feel that this approach is still valuable in advancing understanding of satellite-derived PIC estimates, even without the ideal dataset for a true validation.

Given the constraints of our study, we hope the reviewer can appreciate that our work represents a step towards further research in this area. While it may not be a perfect comparison, we do believe it provides information of interest for the scientific community, and we have always framed our work as exploratory in nature.

We also would like to emphasise that all of these limitations – including the absence of in-situ PIC measurements, the reliance on carbonate in coccoliths, and the impact of cloud cover on satellite data – are clearly highlighted in the manuscript. We are fully transparent about these constraints and it is not our intention to present the results as a comprehensive comparison or to overlook these factors.

Again, I think the much more robust message, which also is more interesting, is that the high signal of PIC seen by satellite south of the polar front is clearly not due to coccolithophores. That is, their data help to define the true polar limits of coccolithophores. They could even present more clearly the sparse coccolithophores they find in those waters. The southern limit is an important observation, and I would say it is the main reason I think the work could be published.

It is true that our data can help to define the limits of coccolithophores in the Antarctic realm (even though we dealt with plankton samples -i.e. time snapshot-), so we added a few sentences in the abstract, discussion and conclusions regarding the southernmost extent of coccolithophores.

Still, we consider this is not the main aim of the paper. We explored the coccolithophore species ecologies in Saavedra-Pellitero et al. (2019) as well as Malinverno et al. (2015), and here we wanted to focus on a different (but related) aspect: the PIC.

We tried to make this clearer in the new version of this manuscript and we even shortened the discussion by deleting most of the section "5.3 *Emiliania huxleyi* morphotypes"

The main concern then is that the manuscript would just be essentially a re-publishing of previous results from the papers cited in Table 1. In that sense, I do think the comparison to satellite work does help in justifying publication, by adding to a new synthesis on defining how we can detect the southern coccolithophore limits. They may consider incorporating data published by other groups (some of which they cite) to support.
Only coccolithophore assemblage data was previously published. All the morphometric measurements and PIC estimates (on top of the satellite data) are new.
The suggestion of compiling published data from other authors to have a larger spatio-temporal span would be a great idea for future work.

The refocus would not justify such a long manuscript, and would need some condensing. Also, the manuscript requires some re-wording of the Title and Abstract, a modest reorganization of the Intro, and a more important reorganization of the Discussion to focus on that point.
Title, abstract and introduction have been modified, and notably shortened.

We reorganised some parts of the results and discussion; both sections are now much more concise.

In summary, I do support eventual publication, but I still think some further revision is needed. I evaluate these as "minor revisions" because I do not think a lot of effort or time would be required, though in fact it would still mean a major change of emphasis.
We hope we were able to provide it with the changes made.

**Detailed comments:**
Title: Still a problem with the title because of the word "versus"
We changed the tile as suggested also by the other reviewer

Abstract, lines 18-19: Again, remove focus on comparison "Here, we combine micropalaeontology and remote sensing to evaluate discrepancies between coccolithophore and satellite-derived PIC in the Pacific SO (in non-bloom conditions)."
This sentence has been removed as suggested also by the other reviewer

Lines 41-43: "This process decreases the alkalinity of surface waters, thereby reducing the uptake of CO2 from the atmosphere into the surface ocean and thus acting in opposition to carbon sequestration by the biological carbon pump (Rost and Riebesell, 2004)." The role of coccolithohores in ocean C uptake is probably more complex. For example, I am not aware that the ballast hypothesis has been completely discarded.
We modified that sentence and added the following one (and a new reference):
"Furthermore, coccolithophores influence the export of PIC and POC to the deep ocean

through the ballasting effects of their coccoliths into the deep sea (e.g. Klaas and Archer, 2002)."

Suggest moving much of the content of the second paragraph of the Intro to much later, so the paper has a focus on better understanding of the "Great Calcite Belt" and coccolithophore penetration of the Southern Ocean, rather than comparing satellite-PIC with coccoltihophores.
We notably modified the Introduction, and brought up the "Great Calcite Belt" earlier in this section.

Lines 106-107: "Our aims are: (1) to estimate the contribution of different coccolithophore taxa and morphotypes to PIC and (2) to compare coccolith-based PIC estimates with satellite-derived PIC values in the Pacific SO." I strongly suggest re-wording aims to match what can actually be achieved with their data. They cannot estimate the contribution of coccolithophores to total PIC without having chemical measurements of total PIC and resolving technical issues with estimating coccolith PIC by microscopy. So these aims must be completely re-written.
Boths aims have been reworded. We hope they are now clear and concise.

Line 125: "Coccolithophores dominate the SO phytoplankton communities" Despite the large number of references cited, I am not comfortable with the word "dominate", especially when considering the entire SO. They are important components of some SO phytoplankton communities, but clearly many SO phytoplankton communities are dominated by diatoms or by other phytoplankton (e.g. Phaeocystis)
We agree with the reviewer, and we reworded that sentence as: "Coccolithophores are important components of some of the SO phytoplankton communities".

Lines 133-139: "and almost always occurs as B morphogroup (types B/C and O). Furthermore, a general southwards decreasing trend in E. huxleyi mass, linked to a latitudinal trend from more calcified E. huxleyi (A morphogroup) to weakly calcified morphotypes (B morphogroup), was already recorded across the Drake Passage (Saavedra-Pellitero et al., 2019)." The concepts and importance of morphogroup and "more calcified"vs "weakly calcified" has not been introduced at this point, so appear out of nowhere. The importance of morphotype should be introduced in a re-organized and condensed introduction.
We added information regarding morphotypes and calcification in the introduction. This implied also adding a couple of new references.

183-184: "The importance of own size measurements for the determination of species-dependent coccolith PIC has been clearly emphasized (Baumann, 2004)". The phrase "own size measurement" does not seem to be correct English. Do the authors mean that it is important to make size measurements on the communities analyzed, rather than assume size measures from the literature?
Yes, we modified this sentence with the suggestion provided by the reviewer

Lines 203-206: "Coccolith volume estimates are likely to contain errors around 40-50% according to Young and Ziveri (2000), so we assumed the largest potential error and added a 50% error bars to our plots, although we note that measuring the actual size range in the

sample can reduce this error to about 5-10% in length and 15-30% in authors' attempt to respond to R1's detailed comments about needing better error analysis. However, I would prefer that the actual error bars refer to the variation in measure perhaps be indicated on the plot perhaps by lines or highlighting (with grey or color) what would be the expected range including this type of error (perhaps showing 15% and briefly how this uncertainty impacts conclusions (focusing on conclusions robust to the uncertainty).

This is indeed a good suggestion. We added a 15% error to (a) in Figures 3 and 4 and checked that both shaded areas (15% and 50%) are colour-blind friendly using https://www.color-blindness.com/coblis-color-blindness-simulator/ We also refer to this in the discussion.

Line 258- Subsection title: "4.1 Coccolith-estimated PIC versus satellite-derived PIC" I strongly recommend moving away from a comparison of one estimation to another in a "versus" sense. Focus on defining the southern boundary of higher E. huxleyi abundances, and note that the satellite estimated PIC measures

We changed the title to: "Coccolith-estimated PIC and satellite-derived PIC" (which is now Section 4.2). As this is something already suggested by Malinverno et al. (2015) and Saavedra-Pellitero et al. (2019), we mention it in the discussion.

Fig. 3-4. I find it difficult to compare parameters with each parameter in a different panel with an offset. I think the figures could be re-organized. For example, one overlay panel (perhaps with right and left y axes) would be total coccolithophores, total detached coccoliths. Another would be E. huxleyi and C. leptoporos abundances. Another would be a column chart showing the relative abundances of the different E. huxleyi morphotypes.

We decided to update Figures 3 and 4 with other suggestions provided by this reviewer, but we did not follow his advice on this specific one.

We wanted to keep the coccolithophore species numbers (visually) separate from the morphometrics, and satellite parameters, so overlaying panels would mean a similar format to all the data. It is kind of a personal preference, but that is the main reason why we kept that part as it was before.

Fig. 5S should be mentioned in the first paragraph of section 4.1. It could even be a main figure after Fig.s 3-4.

We moved that figure from the supplementary material to the manuscript as a main figure (now Figure 7) and mentioned it in section 4.2 (we reorganised the results part, so section 4.1 is now 4.2), but in the second paragraph of that section.

Fig. 2 and Fig. 6. How is tube width measured in the Type A overcalcified coccoliths where the tube overgrows the central area in an irregular manner? How is T element width overcalcified coccoliths where the elements are fused? Where is the data on element width?

This is a very good question. It was very challenging to measure the tube width when it was overgrown in an irregular manner. We aimed for the best possible fitting in the manual measurements (New Zealand samples), and we trusted the Cocobiom 2 macro to automatically do the same.

T elements were measured, when they were fused or it was not possible to measure them for any reason, we indicated it with N/A (not available). Still that data, even if it is available in tables 1S and 2S in the supplementary material.

Fig. 7 and Fig. 8 should be presented in Results.
Previous figure 8 is mentioned in the Results section, but previous 7 (now 9) is only mentioned in the Discussion section.

The Discussion should start with defining the southern boundary, based on the new results here as well as reviewing previously published microscopic evidence.
We re-arranged the results and discussion part.
We deleted the previous section 5.3. and kept just two sections (which we rewrote) in the discussion part to make it more concise and focus on the main topic: the PIC estimates.

Note that several references were deleted while rewriting the introduction and discussion parts.